# Unveiling Topological Structures from Language: A Survey of Topological Data Analysis Applications in NLP

## Abstract

The surge of data available on the Internet has led to the adoption of various computational methods to analyze and extract valuable insights from this wealth of information. Among these, the field of Machine Learning (ML) has thrived by leveraging data to extract meaningful insights. However, ML techniques face notable challenges when dealing with real-world data, often due to issues of imbalance, noise, insufficient labeling, and high dimensionality. To address these limitations, some researchers advocate for the adoption of Topological Data Analysis (TDA), a statistical approach that discerningly captures the intrinsic shape of data despite noise. Despite its potential, TDA has not gained as much traction within the Natural Language Processing (NLP) domain compared to structurally distinct areas like computer vision. Nevertheless, a dedicated community of researchers has been exploring the application of TDA in NLP, yielding **100 papers** we comprehensively survey in this paper. Our findings categorize these efforts into theoretical and non-theoretical approaches. Theoretical approaches aim to explain linguistic phenomena from a topological viewpoint, while non-theoretical approaches merge TDA with ML features, utilizing diverse numerical representation techniques. We conclude by exploring the challenges and unresolved questions that persist in this niche field.

## 1 Introduction

Proliferation of the Internet has given rise to the generation of massive amounts of data. These massive amounts of data when processed can solve many crucial issues plaguing our current society. Due to this well-established notion among stake-holding institutions, the Machine Learning (ML) field has been thriving as a tool that extracts trends and solutions to non-trivial problems. However, real-world data tends to be noisy, heterogeneous, imbalanced, have missing labels, contain high-dimensionality, etc., often making the adoption of ML techniques to such datasets non-trivial. Therefore, to extract meaningful findings from data, specifically real-world data, clever techniques that extract additional features, while *preserving the structure of the data need to be employed*. To that end, a small niche community for *Topological Data Analysis (TDA) applications in NLP* has emerged. Being promised as a technique that can extract and analyze the shape/topology of data, TDA has great potential in mitigating such issues witnessed in real-world data. Thus by applying TDA to NLP, we obtain "*topological structures from language*," which refers not to intrinsic properties of raw text itself, but to the structures that emerge when linguistic data is mapped into high-dimensional embedding spaces. These induced topologies capture relationships among words, sentences, or documents based on their learned representations, rather than any inherent topological features of the text. In addition, the phrase also helps us distinguish the application of TDA and other topological approaches applied to other fields, such as computer vision from NLP.

TDA is a "collection of powerful tools that can quantify shape and structure in data"[1] and is inspired by the algebraic topology and geometry mathematical fields. The benefits of TDA are vast, including the ability to extract additional features, typically not captured by other feature extraction techniques (Uchendu et al., 2024; Michel et al., 2017; Papamarkou et al., 2024). These features are known as *topological features*. Unsurprisingly, since TDA is used to capture topological features, it has been applied to many tasks where data has distinct graphical structures (Papamarkou et al., 2024; Hensel et al., 2021). These include tasks that have obvious graph-like structures, such as protein classification

---

[1]https://www.indicative.com/resource/topological-data-analysis/

Figure 1: Number of NLP papers using TDA published each year from 2012 to July-2025

(Dey & Mandal, 2018; Lamine et al., 2023; Valeriani et al., 2024) and drug discovery (Alagappan et al., 2016); to those that are not so obvious such as diabetes classification (Wamil et al., 2023; Skaf & Laubenbacher, 2022), image classification (Horn et al., 2022; Trofimov et al., 2023), and time series analysis (Petri & Leitao, 2020; Tymochko et al., 2021; Gholizadeh & Zadrozny, 2018). However, since the shape of a text is not apparent, it has not gained as much attention in Natural Language Processing (NLP) as it has done in the Computer Vision (Horn et al., 2022; Trofimov et al., 2023) and Medical (Singh et al., 2023; Nielson et al., 2015) domains. Still, several researchers have found ways to extract unique features using TDA, in which other typical numerical representation techniques in text, such as TF-IDF, Word2Vec embeddings, BERT embeddings, etc., cannot extract.

Furthermore, while these standard numerical representations can perform well in classifying many benchmark datasets, they are not always robust to more realistic datasets that tend to have harder constraints, such as heterogeneity of labels, imbalanced, noisy, missing data, unlabeled data, etc. Since one of the benefits of TDA is to find the features after deformation techniques have been applied to the data, it tends to be more robust to these harder constraints. Therefore, for this reason, the small niche *TDA applications in NLP* community has explored several ways to extract additional features for classification, interpreting/explaining model performance, and explaining linguistic phenomena within text or speech. The first application of TDA in NLP was published in 2012 (Wagner et al., 2012), and since then there have been over 90 papers applying TDA in NLP. See Figure 1 for the gradual acceleration in the number of published works in TDA applications on NLP tasks, and we project that this trend will continue in the future. All these papers will be discussed in the survey.

TDA aims to answer the main question - *what is the true shape of a data?* We survey 100 papers that have attempted to find an answer through various approaches. We categorize all observed approaches that incorporate TDA in NLP into two categories, namely *theoretical* (Karlgren et al., 2014; Port et al., 2018) and *non-theoretical* (Zhu, 2013; Doshi & Zadrozny, 2018) approaches. *Theoretical* approaches involve using TDA to explain linguistic phenomena by probing the topological space, shape, and evolution of topics. On the other hand, *Non-theoretical* approaches mainly discuss how to effectively apply existing numerical representation techniques in NLP to extract novel topological features with TDA. We will first discuss the principles behind TDA and the two main techniques employed for TDA feature extraction: *Persistent Homology* and *Mapper*.

| Alternative Method | Description |
|---|---|
| Principal Component Analysis (PCA) | Reduces dimensionality while preserving variance using orthogonal transformations. |
| t-Distributed Stochastic Neighbor Embedding (t-SNE) | Projects high-dimensional data into 2D or 3D space while preserving local relationships. |
| Uniform Manifold Approximation and Projection (UMAP) | Similar to t-SNE but faster and better at preserving global structure. |
| Multidimensional Scaling (MDS) | Projects high-dimensional data into lower dimensions by preserving pairwise distances. |
| Autoencoders | A type of Neural network that learns compressed representations of data via encoding-decoding processes. |
| Manifold Learning (Isomap, LLE, etc.) | Captures intrinsic structures in high-dimensional data through graph-based techniques. |
| Graph-based Learning (Spectral Clustering, GNNs) | Uses graphs to model relationships and structure within data. |
| Clustering Methods (DBSCAN, K-Means, HDBSCAN) | Groups similar data points based on distance and density. |
| Kernel Methods (SVM, Kernel PCA) | Uses non-linear mappings to extract complex structures in data. |
| Geometric Deep Learning | Uses neural networks on non-Euclidean spaces like graphs and manifolds. |
| Geometric techniques (Delaunay triangulation, Convex hull) | Uses geometrical techniques to extract characteristics of data by analyzing spatial boundaries |

Table 1: Alternatives to Topological Data Analysis in NLP

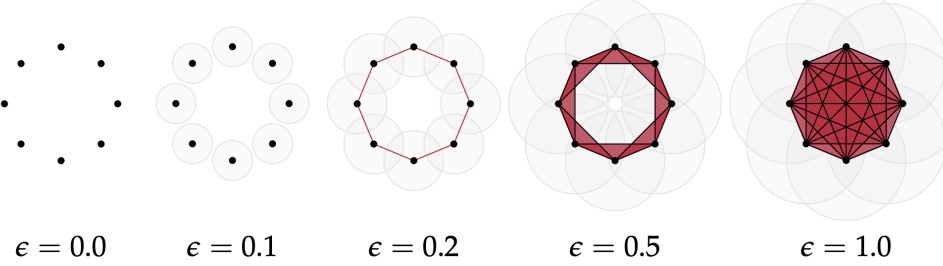

$\epsilon = 0.0$     $\epsilon = 0.1$     $\epsilon = 0.2$     $\epsilon = 0.5$     $\epsilon = 1.0$

Figure 2: Illustration of the Persistent Homology technique using different radii to find the persistent features (Rieck, 2020). $\epsilon$ is the ball diameter.

## 2 Topological Data Analysis (TDA)

Topology is defined as *"the study of geometric properties and spatial relations unaffected by the continuous change of shape or size of figures,"* (Oxford Dictionary). TDA is then a collection of powerful techniques that can quantify the shape and structure of data (Munch, 2017). While there are alternatives to TDA (i.e., Table 1), TDA is the only technique that can extract not only local but global features, preserving the shape and structure of the data, and is also robust to insufficient data. Two main techniques are used to extract TDA features: *Persistent Homology* and *Mapper*.

### 2.1 Persistent Homology

Persistent Homology (PH) (Edelsbrunner et al., 2008) is the most popular TDA technique. It uses algebraic topology methods to extract topological signatures at different spatial dimensions. This process involves representing data as a point cloud and performing deformation or perturbation processes to extract the true "shape" of data after the noise

has been removed. To achieve this, PH employs Vietoris-Rips complex (Munch, 2017). Vietoris-Rips complex is a way to build simplicial complexes which are used to represent data in a topological space. A simplicial complex is a topological space built by putting points, lines, and higher dimensional shapes together. These formations reveal features that are holes in different dimensions, represented as *betti numbers* ($\beta_d$, $d$-dimension). Holes in the 0-dimension ($\beta_0$) is represented as one vertex, 1-dimension ($\beta_1$) is represented as an edge, 2-dimension ($\beta_2$) is represented as a triangle. Further, these features are called connected components, loops/tunnels, and voids, respectively.

Using the method described above, data is represented as a point cloud, and circles are drawn around each point. Next, the radius of each circle is increased using a defined range of points, such that if the circles get bigger and touch, one of the points disappears and this is recorded as a $death$. Additionally, this process of perturbation in different dimensions can cause the $birth$ of a new hole which is also recorded. Thus, due to these deformations, the following TDA features can be extracted and recorded in a 3-column matrix, which consists of columns representing - the $birth$ (formation of holes), $death$ (deformation or the closing of holes), and persistence features. Persistence is defined as the length of time it took a feature to disappear or die ($death - birth$). The $death$ is recorded with the radii value at which the points overlap. Lastly, TDA features are typically visualized in a persistence diagram which is a visual representation of the 3-column matrix of TDA features. Figure 2 illustrates an example of the process of extracting TDA features using Persistent Homology. Other ways of visualizing TDA features include persistence images (Adams et al., 2017) and barcode plots (Ghrist, 2008). In terms of application, PH has been used to extract novel features to complement existing NLP representations and improve various classification performances (Doshi & Zadrozny, 2018; Uchendu et al., 2024; Wu et al., 2022).

> **Persistent Homology.** *This is a TDA technique that studies the deformation of "holes" in different dimensions. Using PH, we can track when features appear and disappear and visualize these features, usually in a persistence diagram. This process allows us to find the true structure of data, typically devoid of noise.*

## 2.2 Mapper

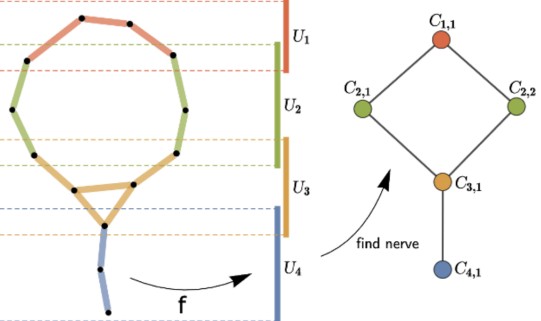

Figure 3: Illustration of Mapper from Murugan & Robertson (2019). The filter function $f$ is a height function, which is a projection onto the y-axis. The cover of the projected space is the four intervals $U_i$. The Mapper graph on the right is a result of applying the rest of the Mapper algorithm and clustering each preimage in the nearest neighbor.

Mapper is a dimension reduction clustering technique for visualizing TDA-extracted topological structures/signatures. It was proposed by Singh et al. (2007) and has been used extensively to visualize topological structures in data to create visually pleasing figures, as well as interpret model performance through data probing (Carlsson, 2020). The Mapper algorithm works in four steps[2] (Figure 3) following Murugan & Robertson (2019)'s instructions: (1) Transform the data to a lower-dimensional space using a filter function $f$, also known as a lens. This implies projecting from one space to another. Options for filter functions include PCA (Maćkiewicz & Ratajczak, 1993), UMAP (McInnes et al., 2018), and any other dimension-reduction algorithms; (2) Create a cover $(U_i)_{i \in I}$ for the projected space, which is typically composed of overlapping intervals with a constant length; (3) Cluster the points in the preimage $f^{-1}(U_i)$ into sets $C_{i,1}, \ldots, C_{i,k_i}$

per interval $U_i$; (4) Create a graph where each vertex represents a cluster set. There is an edge between two vertices if the corresponding clusters share common points. Points in the same neighborhood are clustered using a defined clustering technique, such as DBSCAN (Ester et al., 1996) to change a cluster of several points into a node of a graph.

The intrinsic nature of the Mapper algorithm makes it advantageous in preserving structure, even with mapping from one dimension to another. Furthermore, the clustering techniques allow it to be used to explain model performance as the clusters and colors have meaning that can be further explored. Finally, Mapper is more useful for exploratory data

---

[2]https://www.quantmetry.com/blog/topological-data-analysis-with-mapper/

analysis, while PH is more useful for analyzing point clouds and examining the persistence of features. In this survey, we will discuss how several researchers use Mapper to explain or enhance several phenomena in NLP tasks.

> **Mapper.** *This is a TDA technique that visualizes the graphical representation of data in order to capture the intrinsic structure. It is very useful for preserving data structure and creating visually pleasing plots which can be investigated manually to find insights.*

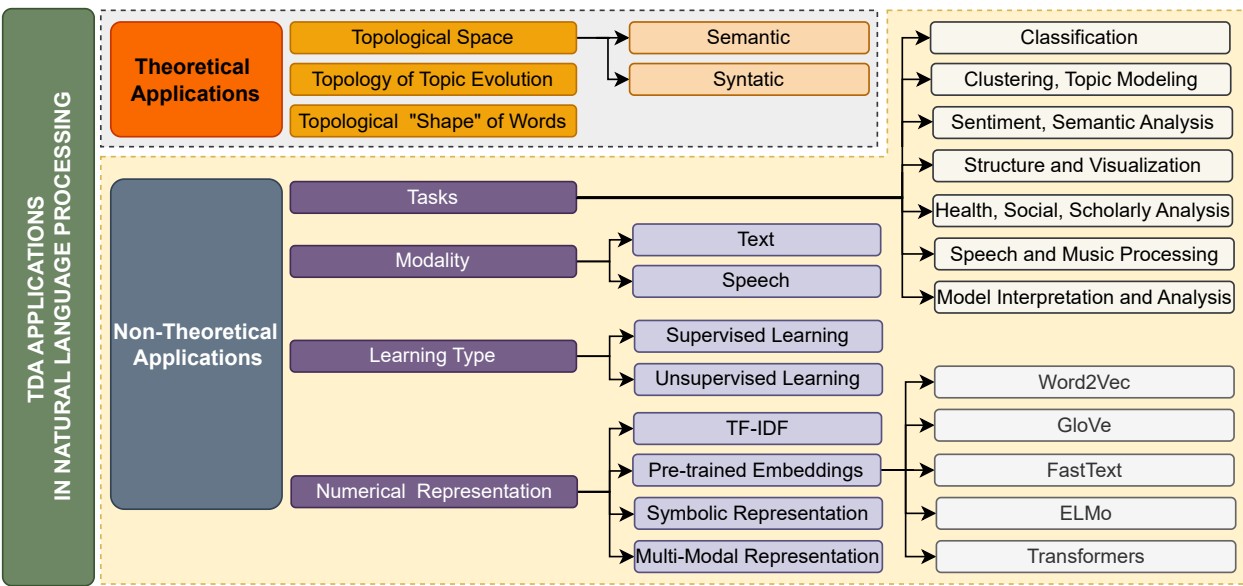

Figure 4: Taxonomy of Topological Data Analysis (TDA) for Natural Language Processing (NLP) Applications

# 3 Selection Criteria for papers surveyed and Taxonomy Development

## 3.1 Selection Criteria

In order to find all NLP papers which applied TDA, we manually searched on Google Scholar using key terms such as *text mining persistent homology*, checking related articles of the relevant papers, their cited papers, and different combinations of all three methods. After, obtaining over 60 papers initially, we started creating a taxonomy and categorizing the papers. Initially, we focused on TDA applications in textual data, but as we searched, we found several applications in speech, and collected such papers. Finally, we removed papers that did not apply TDA to text or human speech data. Papers removed either applied TDA to a graphical representation of reddit social networks, applied non-TDA topological techniques, or applied TDA to non-speech audio data. Using these criteria we selected only papers that fit and collected the rest following the schema.

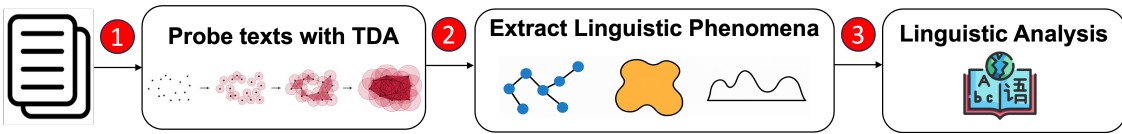

Figure 5: Illustration of the theoretical approaches researchers have employed to (1) probe texts, (2) extract TDA features, (3) use these features to explain or confirm known linguistic phenomena.

| Name | Category | Task | TDA Technique |
|------|----------|------|---------------|
| (Karlgren et al., 2014) | Topological Space (Semantic) | Identify topical density of the space | Mapper |
| (Cavaliere et al., 2017) | Topological Space (Semantic) | Extracts main concepts from text | Persistent Homology |
| (Wagner et al., 2012) | Topological Space (Semantic) | Analyzes document similarities | Persistent Homology |
| (Port et al., 2018) | Topological Space (Syntactic) | Analyzes syntactic parameters of different language families | Persistent Homology |
| (Port et al., 2022) | Topological Space (Syntactic) | Explains linguistic structures with homoplasy phenomena | Persistent Homology |
| (Sami & Farrahi, 2017) | Topology of Topic Evolution | Topic evolution within documents | Persistent Homology |
| (Draganov & Skiena, 2024) | Topological "Shape" of Words | Investigates the "shape" of language phylogenies in the Indo-European language family | Persistent Homology |
| (Fitz, 2022) | Topological "Shape" of Words | Captures grammatical structure expressed by corpus using *word manifold* | Persistent Homology |
| (Fitz et al., 2024) | Topological "Shape" of Words | Measures the topological complexity of Transformer-based hidden representations | Persistent Homology |
| (Dong, 2024) | Topological "Shape" of Words | Investigates the topological shapes of South American languages—Nuclear-Macro-Jê (NMJ) and Quechuan families | Persistent Homology |
| (Bouazzaoui et al., 2021) | Topological "Shape" of Words | Investigates the similarity between the topological shapes of the Tifinagh and Phoenician scripts | Persistent Homology |

Table 2: Theoretical Applications of TDA in NLP

## 3.2 Taxonomy Development

Based on the papers selected for the survey, we were able to categorize the applications of these papers into two approaches - *theoretical* and *non-theoretical* applications. See description below:

- **Theoretical applications of TDA in NLP**: These focus on understanding, characterizing, or proving properties of language and its representations through the lens of topology. They are less about immediate performance gains and more about insight. This application aims to answer the question - "What do the shapes of embedding spaces tell us about language itself and our models of it?" Example - Analyzing embedding spaces: Using persistent homology to study whether semantic clusters, or syntactic structures, correspond to stable topological features, and finding out what that tells us about language.
- **Non-theoretical (practical) applications of TDA in NLP**: These treat TDA as a tool for solving tasks, regardless of whether deeper linguistic/topological insights are obtained. The emphasis is on utility. This application aims to answer the question - "How can topological summaries directly help with applied NLP tasks?" Example - Feature engineering: Feeding persistence diagrams or topological signatures into classifiers for sentiment analysis, topic detection, or authorship attribution.

# 4 Theoretical Approaches of TDA in NLP

Since the field of NLP is very interested in representing and analyzing texts or speech in meaningful ways, several theoretical approaches have been proposed to investigate how well these approaches align with linguistic principles. Thus to explain or confirm linguistic phenomena within the NLP paradigm, a few researchers have proposed topological approaches for probing NLP techniques. See Figure 5 for an illustration of this pipeline. By employing TDA techniques - Persistent homology or Mapper to probe for linguistic phenomena, researchers aim to capture the *topological space* (both *semantic* and *syntactic* relationships) of texts, analyze and visualize the *topology of topic evolution* within texts, and extract the *topological shape* of words. See Table 2 for the theoretical approaches. In essence, these theoretical topological methods provide a conceptual bridge between linguistic theory and mathematical topology.

> **Theoretical vs. Non-Theoretical approaches.** *The main difference between the theoretical and non-theoretical approaches in this context is - Theoretical approaches use TDA (sometimes in combination with other numerical techniques) to understand and explain linguistic phenomena, while Non-theoretical approaches use TDA to enhance or explain model performance. This means that the Theoretical approaches are focused on linguistic phenomena, while Non-theoretical approaches are focused on model performance. Thus, using linguistic phenomena understood from the Theoretical approaches, can inform and enhance Non-theoretical approaches.*

## 4.1 Topological Space

### 4.1.1 Semantic Topological Space

A **semantic topological space** is a conceptual framework used to represent and analyze the relationships between the meanings (semantics) of words, phrases, or other linguistic units in a topological or shape structure. This representation involves mapping these units into a mathematical space where the distance or structure between them reflects semantic similarity or other relationships (i.e., Euclidean space $\rightarrow$ Topological space).

Karlgren et al. (2014) visualizes the topological semantic space of text using Mapper which identifies the topical density of the space. To capture topological properties, they train two semantic spaces in a specific topical domain (Karlgren et al., 2014). One space was trained only on articles of similar topics, and the other on introductory paragraphs of those same articles. Findings reveal that clusters of main concepts remained close for the space trained only on articles of similar topics. For the other topological space, the main concepts were randomly distributed (Karlgren et al., 2014). This suggests that richer and denser data can be used to capture the semantic topological space better than sparser data.

In addition, Cavaliere et al. (2017) extracts main concepts from the texts by probing the context-aware semantic topological space built with simplicial complexes. Finally, Wagner et al. (2012) uses TF-IDF to numerically represent the top 10-50 words in a corpus and build a topological space that analyzes the structure of similarities within several documents. This topological space is built using discrete Morse theory and Persistent Homology to find meaningful topological patterns (Wagner et al., 2012). However, as of 2012, they found that their technique was unsuccessful due to the computational costs, which is a testament to how the NLP field has improved such that we now have more tractable solutions, such as dimensionality reduction algorithms (McInnes et al., 2018), and TDA packages (Ripser (Bauer, 2021), Sklearn-TDA (Saul & Tralie, 2019), PHAT (Bauer et al., 2017), pytorch-topological[3]), as well as compute resources to efficiently construct topological spaces from large or complex data.

> **Insight (Semantic Space).** *The semantic topological space is explored by researchers to identify semantic linguistic principles captured in texts through a topological lens. Most of the applications in this section involve understanding the semantic similarity between text pairs from a topological lens.*

### 4.1.2 Syntactic Topological Space

A **syntactic topological space** is a theoretical framework used to represent and analyze the relationships between syntactic structures from a topological perspective. This concept is particularly relevant in linguistics, where it helps model and understand the structural aspects of language such as grammar, sentence construction, or the hierarchical organization of those linguistic units.

Therefore, Port et al. (2018) analyzes how syntactic parameters are distributed over different language families, including Indo-European, Niger-Congo, Austronesian, and Afro-Asiatic families. For instance, features in $\beta_0$ capture the subdivision into historical, and features in $\beta_1$ capture syntactic differences between branches of families of languages, as well as the syntactic influences between them (Port et al., 2018). They investigate the syntactic topological structures of language families, specifically Indo-European, Niger-Congo, Austronesian, and Afro-Asiatic families. Port et al. (2018) show that the three persistent connected components ($\beta_0$) in the Niger-Congo family represents its

---

[3]https://github.com/aidos-lab/pytorch-topological

three subfamilies - Mande, Atlantic-Congo, and Kordofania. The syntactic topological structures of these languages also reveal the historical linguistic phenomena that the Hellenic branch played a role in the historical development of the Indo-European languages (Port et al., 2018).

Similarly, Port et al. (2022) probes the interpretability of the syntactic topological space. This is done by incorporating more explanation of linguistic structures by introducing so-called homoplasy phenomena to explain persistent loops further. Homoplasy phenomena in syntax are observed when dissimilar languages exhibit syntactic similarities (Port et al., 2022). Findings reveal that the Indo-European family languages - Czech, Lithuanian, Middle Dutch, and Swiss German have the same homoplasy phenomena (Port et al., 2022). Using this phenomenon to explain the appearance of persistent loops only when the 4 languages are present makes sense. This is because Middle Dutch and Swiss German are similar, however, Czech and Lithuanian are so different from them, making the homoplasy phenomenon the most reasonable explanation (Port et al., 2022).

> **Insight (Syntactic Space).** *The syntactic topological space captures the syntactic structure of language (i.e., grammar, etc.) from a topological lens. Using this framework, researchers confirm linguistic phenomena in language families and subfamilies by exploring the syntactic relationship between languages. Thus, a novel application of this framework could include the discovery of new linguistic phenomena within syntactic structures.*

### 4.2  Topology of Topic Evolution

The **topology of topic evolution** refers to the study and representation of how topics, themes, or concepts develop and change over time within a given corpus of texts or discourses in a topological space/structure. This concept is particularly relevant in fields where understanding the temporal dynamics of topics can provide insights into trends, shifts in public opinion, or the development of scientific or cultural themes.

Thus, Sami & Farrahi (2017) utilizes TDA to visualize the relationship between words in a text block, words in a corpus, and text blocks in a corpus. Text blocks represent a chapter/section in a book, a document in a media corpus, and a webpage in a web corpus (Sami & Farrahi, 2017). They visualize both local context (i.e., each text block in a set of sentences), and global context (i.e., occurrence of extracted words in the corpus) features. These features are extracted by using the circular topology to represent words. Then, the peripheral nature of the text block and corpus can be visualized using these features. With the Local context features, dimension reduction is achieved by stemming the prefixes and suffixes of words. While, for the Global context features, word movement is captured, which analyzes topic evolution. Finally, findings reveal that using the circular topology in 2D space, core words from the corpus stay close to the center, and the explanatory words remain close to the circle's periphery.

> **Insight (Topic Evolution).** *Exploring the topology of topic evolution is a novel framework for capturing the topology of topics in a corpus. The findings suggest that this framework can be adopted to evaluate the utility of a summarization, paraphrasing, or obfuscating model, by comparing the topology of the topic evolution of the original vs. the perturbed texts.*

### 4.3  Topological "Shape" of Words

The **topological "shape" of words** is a conceptual framework in linguistics and cognitive science that explores the structural properties of *words*. This framework leverages ideas from topology to capture the true shape of words in a linguistically meaningful way. Thus, using topological methods such as TDA, the structural properties of words can be extracted and analyzed.

Draganov & Skiena (2024) captures the "shape" of words for several languages by comparing the phylogenies or evolutionary history of language in the Indo-European language family. Initially, numerically representing the texts with FastText, they use persistent homology to construct language phylogenetic trees for over 81 Indo-European languages. Experiments reveal that: (1) the shape of the word embedding of a language carries historical and structural information, similar to Port et al. (2018; 2022)'s findings; and (2) TDA methods can successfully capture aspects of the shape of language embeddings (Draganov & Skiena, 2024). Next, Fitz (2022) introduces a novel terminology - *word*

*manifold*, which is a simplicial complex, whose topological space captures grammatical structure expressed by the corpus. This is done by implementing a technique for generating topological structure directly from strings of words. Experiments reveal that the homotopy type of the word manifold is also influenced by linguistic structure (Fitz, 2022). Similarly, Fitz et al. (2024) measures the topological complexity (known as *perforation*) of the hidden representation of Transformer-based models to understand their topological shapes.

Similar to Port et al. (2018; 2022) and Draganov & Skiena (2024), Dong (2024) extracts the topological shapes of languages. Specifically South American languages - the Nuclear-Macro-Jê (NMJ) and Quechuan families using TDA. By using techniques like multiple correspondence analysis (MCA) for dimension reduction of the categorical-valued dataset and persistent homology, Dong (2024) visualizes each language in the selected families as a point cloud. This forms the topological shape of the South American languages, such that languages close together are more similar. By comparing the topological shapes of the languages, it is observed that there are major distinctions between the Jê-proper and the non-Jê-proper languages, as well as the northern and southern Quechuan languages (Dong, 2024). Finally, Bouazzaoui et al. (2021) explores the topological similarity of the shapes of two writing systems - Tifinagh and Phoenician scripts.

> **Insight (Shape of Words).** *The topological "shape" of words is a concept that has interested several linguists, as it can be used to confirm and discover linguistic phenomena within languages. It is the most studied theoretical approach, with the main application focused on capturing the shape of several languages. This concept combines all other frameworks like the semantic and syntactic topological spaces to capture a linguistically informed topological shape of texts.*

## 5 Non-theoretical Approaches of TDA in NLP

There are several ways to categorize the applied/non-theoretical TDA applications in NLP. These applications can be categorized by *task*, *learning type*, *modality*, *TDA technique*, and *numerical representation*. We observe that categorizing these TDA applications by task and numerical representation is more meaningful than the other categories since those categories are binary and not very descriptive of the landscape. Out of these dimensions, the numerical representation showcases the bottleneck for extracting useful TDA features. See Figure 6 for an illustration of the pipeline for extracting TDA features from numerically represented texts. Therefore, while we focus on both task and numerical representation, our main taxonomy for the non-theoretical applications is centered on how TDA features are extracted from different forms of numerical representations. Figure 4 illustrates the taxonomy of non-theoretical applications of TDA in NLP tasks. See Table 3 for the list of non-theoretical approaches.

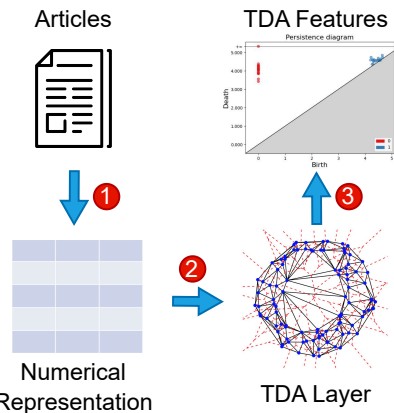

Figure 6: Illustration (inspired by Uchendu et al. (2024)) of the Non-theoretical approach of using TDA as a feature extractor in NLP with three steps: (1)-extracting numerical representations, (2)-reformatting for TDA's inputs, and (3)-extracting TDA features.

### 5.1 Tasks

We categorize the NLP tasks to which TDA has been applied into seven categories:

1. **Classification**: The most popular application is deepfake text detection (Løvlie, 2023; Tulchinskii et al., 2024; Kushnareva et al., 2024; 2021; Uchendu et al., 2024; Wei et al., 2025).
2. **Clustering and Topic Modeling**: The most popular application is document clustering and topic modeling (Holmes, 2020; Guan et al., 2016).

| Name | Task | Problem | Numerical Representation | Learning Type | Modality | TDA |
|------|------|---------|--------------------------|---------------|----------|-----|
| SIFT (Zhu, 2013) | cl | child vs. adolescent writing detection | TF-IDF | Supervised | Text | PH |
| (Løvlie, 2023) | cl | deepfake text detection | TF-IDF, GloVe | Supervised | Text | PH |
| (Huang, 2022) | cl | election speech feature extraction | TF-IDF | Supervised | Text | PH |
| (Doshi & Zadrozny, 2018; Shin, 2019) | cl | movie genre cl | TF-IDF | Supervised | Text | PH |
| (Sovdat, 2016) | cl | distinguishing between languages | TF-IDF | Supervised | Text | PH |
| (Savle et al., 2019) | S & SA | legal document entailment | TF-IDF | Supervised | Text | PH |
| (Michel et al., 2017) | S & SA | clustering and sentiment analysis | TF-IDF, GloVe | Supervised | Text | PH |
| DoCollapse (Guan et al., 2016) | C & TM | keyphrase extraction | TF-IDF | Unsupervised | Text | PH |
| TOPOL (Torres-Tramón et al., 2015) | C & TM | Twitter topic detection | TF-IDF | Unsupervised | Text | PH |
| (Kumar & Sarkar, 2022) | C & TM | text summarization | TF-IDF | Unsupervised | Text | PH |
| (van Veen, 2020) | cl | Propaganda tweet cl | TF-IDF | Unsupervised | Text | M |
| (Elyasi & Moghadam, 2019) | cl | classify Persian poems | TF-IDF | Supervised | Text | PH & M |
| (Effah, 2017) | cl | age group categorization of lonely people | TF-IDF | Supervised | Text | PH & M |
| (Maadarani & Hajra, 2020) | cl | nursery rhyme classification from different continents - Australia, Asia, Africa, Europe, and North America | TF-IDF | Supervised | Text | PH |
| (Haghighatkhah et al., 2022) | S & V | building document structure | Pre-trained (Word2Vec) | Unsupervised | Text | PH |
| BERT+TDA (Wu et al., 2022) | S & SA | contradiction detection | Pre-trained (Word2Vec) | Supervised | Text | PH |
| (Yessenbayev & Kozhirbayev, 2022) | MI & A | speaker recognition & text processing | Pre-trained (Word2Vec) | Unsupervised | Speech | PH |
| (Yessenbayev & Kozhirbayev, 2024) | MI & A | speaker recognition & text processing | Pre-trained (Word2Vec) | Unsupervised | Speech | PH |
| (Cornell, 2020) | S & SA | building a topological search engine | Pre-trained (Word2Vec) | Unsupervised | Text | M |
| (Holmes, 2020) | C & TM | document clustering and topic modeling tasks | Pre-trained (Word2Vec) | Supervised | Text | M |
| (Rawson et al., 2022) | S & SA | word sense induction and disambiguation | Pre-trained (Word2Vec) | Unsupervised | Text | PH |
| (Temčinas, 2018) | S & SA | word sense induction and disambiguation | Pre-trained (Word2Vec, GloVe) | Unsupervised | Text | PH |
| (Feng et al., 2024) | MI & A | Geometry of textual data augmentation | Pre-trained (Word2Vec) | Supervised | Text | PH |
| (Tymochko et al., 2021) | cl | fraudulent paper detection | Pre-trained (Word2Vec, GloVe, ElMo) | Supervised | Text | PH |
| (Petri & Leitao, 2020) | H,S,& SA | disease epidemic prediction | Pre-trained (Word2Vec) | Supervised | Text | PH |
| (Tymochko et al., 2020) | cl | finding topological loops in logical statements | Pre-trained (Word2Vec) | Unsupervised | Text | PH |
| (Wright & Zheng, 2020) | C & TM | distinguish subsets in data | Pre-trained (Word2Vec) | Unsupervised | Text | PH |
| (Paluzo Hidalgo et al., 2019) | S & SA | measuring the distance between the literary style of Spanish poets | Pre-trained (Word2Vec) | Supervised | Text | PH |
| (Bailey & Heiligman, 2025) | S & SA | detecting narrative shifts | Pre-trained (Word2Vec) | Unsupervised | Text | PH |
| (Gholizadeh et al., 2018) | cl | extract the topological signatures of novelists | Pre-trained (GloVe) | Supervised | Text | PH |
| TIES (Gholizadeh et al., 2020) | S & SA | document categorization & sentiment analysis | Pre-trained (GloVe) | Unsupervised | Text | PH |
| (Spannaus et al., 2024) | MI & A | phenotype prediction and news group categorization | Pre-trained (GloVe) | Unsupervised | Text | M |
| (Zadrozny, 2021b) | S & SA | finding topological loops in logical statements | Pre-trained (GloVe) | Unsupervised | Text | PH |
| (Byers, 2021) | H,S,& SA | social anxiety detection | Pre-trained (GloVe) | Supervised | Text | PH |
| (Haim Meirom & Bobrowski, 2022) | MI & A | compare cross-lingual sentence representations | Pre-trained (GloVe) | Unsupervised | Text | PH |
| (Zadrozny, 2021a) | MI & A | investigates the manifestations of intelligence and understanding in neural networks | Pre-trained (GloVe) | Supervised | Text | PH |
| (Deng & Duzhin, 2022) | cl | fake news detection | Pre-trained (GloVe, BERT) | Supervised | Text | PH |
| (Novak, 2019) | H,S,& SA | analyzing scholarly network | Pre-trained (GloVe, BERT) | Unsupervised | Text | PH |
| (Jakubowski et al., 2020) | S & SA | (1) polysemy word classification, and (2) word sense induction & disambiguation | Pre-trained (FastText) | Unsupervised | Text | PH |
| (Triki, 2021; Shehu, 2024) | S & SA | polysemy word cl | Pre-trained (FastText) | Supervised | Text | PH |
| PHD (Tulchinskii et al., 2024) | cl | deepfake text detection | Pre-trained (Transformers - CLS) | Supervised | Text | PH |
| Short-PHD (Wei et al., 2025) | cl | deepfake text detection (for short-text) | Pre-trained (Transformers - CLS) | Supervised | Text | PH |
| (Guilinger et al., 2025) | cl | deepfake text detection (for academic abstracts) | Pre-trained (Transformers - CLS) | Supervised | Text | PH |
| (Kushnareva et al., 2024) | cl | deepfake text detection | Pre-trained (Transformers - CLS) | Supervised | Text | PH |
| (Gourgoulias et al., 2024) | MI & A | class separability estimation | Pre-trained (Transformers - CLS) | Unsupervised | Text | PH |
| TopoBERT (Rathore et al., 2023) | S & V and MI & A | visually analyzing the fine-tuning process of a Transformer-based model | Pre-trained (Transformers - CLS) | Unsupervised | Text | M |
| (Lavery et al., 2024) | cl | fake news detection | Pre-trained (Transformers - CLS) | Supervised | Text | PH |
| (Das et al., 2021) | cl | classification of public speaking ratings from TED talks | Pre-trained (Transformers - CLS) | Supervised | Text | PH |
| (Byrne et al., 2022) | C & TM | topic modeling | Pre-trained (Transformers - CLS) | Unsupervised | Text | M |
| TOPFORMER (Uchendu et al., 2024) | cl | deepfake text detection | Pre-trained (Transformers - Hidden) | Supervised | Text | PH |
| (García, 2022) | S & SA | polysemy word cl | Pre-trained (Transformers - Hidden) | Supervised | Text | M |
| (Bensalem et al., 2025) | S & SA | polysemy word cl | Pre-trained (Transformers - Hidden) | Unsupervised | Text | PH |
| (Alexander & Wang, 2023) | H,S,& SA and S & V | Hate speech, Misinformation & Psychiatric disorder cl | Pre-trained (Transformers - Hidden) | Supervised | Text | M |
| Persistent BERT Compression and Explainability (PBCE) (Balderas et al., 2025) | MI & A | Model compression | Pre-trained (Transformers - Hidden) | Unsupervised | Text | PH |
| Persistent Similarity (Gardinazzi et al., 2024) | MI & A | Probing layers in LLMs | Pre-trained (Transformers - Hidden) | Unsupervised | Text | PH |
| (Sun & Nelson, 2023) | MI & A | Correlation between sentence vectors | Pre-trained (Word2Vec, Transformers - Hidden) | Unsupervised | Text | PH |
| Persistence Scoring Function (Chauhan & Kaul, 2022) | MI & A | captures the homology of the high-dimensional hidden representations | Pre-trained (Transformers - Hidden) | Unsupervised | Text | PH |
| Topological Densification (García-Castellanos et al., 2024) | MI & A | zero-shot model stitching | Pre-trained (Transformers - Hidden) | Unsupervised | Text | PH |
| (Fitz, 2023) | MI & A | topology of fairness | Pre-trained (Transformers - Hidden) | Unsupervised | Text | M |
| (Fay et al., 2025) | MI & A | adversarial vs. non-adversarial text representation in LLMs | Pre-trained (Transformers - Hidden) | Unsupervised | Text | PH |
| (Ruppik et al., 2024) | C & TM | dialogue term extraction | Pre-trained (Transformers - Hidden) | Supervised | Text | PH |
| (Kudriashov et al., 2024) | MI & A | polypersonality | Pre-trained (Transformers - Hidden) | Unsupervised | Text | PH |
| (Bazarova et al., 2025) | cl | LLM Hallucination detection | Pre-trained (Transformers - Attention) | Supervised | Text | PH |
| (Kushnareva et al., 2021) | cl | deepfake text detection | Pre-trained (Transformers - Attention) | Supervised | Text | PH |
| (Cherniavskii et al., 2022; Proskurina et al., 2023; Jain et al., 2024) | S & SA | grammatical acceptability judgment | Pre-trained (Transformers - Attention) | Supervised | Text | PH |
| (Kostenok et al., 2023) | MI & A | Uncertainty estimation of model predictions | Pre-trained (Transformers - Attention) | Supervised | Text | PH |
| (Perez & Reinauer, 2022) | cl | spam detection, grammatical acceptability judgment, and movie sentiment analysis | Pre-trained (Transformers - Attention) | Supervised | Text | PH |
| (Sakurai et al., 2025) | cl | Authorship attribution of Japanese texts | Pre-trained (Transformers - Attention) | Supervised | Text | PH |
| (Pollano et al., 2024) | cl | out-of-distribution detection | Pre-trained (Transformers - Attention, CLS) | Supervised | Text | PH |
| (Proskura & Zaytsev, 2024) | MI & A | estimation of weights for ensembles of classification models | Pre-trained (Transformers - Attention, CLS) | Supervised | Text | PH |
| (Arun et al., 2025) | S & V and S & SA | controversial vs. non-controversial political discourse detection | Pre-trained (Transformers - CLS) | Supervised | Text | PH |
| (Snopov & Golubinskiy, 2024) | cl | vulnerability detection in code | Pre-trained (Transformers - Attention) | Supervised | Text | PH |
| TopoHuBERT (Tulchinskii et al., 2023) | S & MP | speaker recognition | Pre-trained (Transformers - Attention) | Supervised | Speech | PH |
| (Vukovic et al., 2022) | S & SA | dialogue term extraction | Pre-trained (Transformers - Attention) | Supervised | Text | PH |
| (Yurchuk & Gurnik, 2023) | S & SA | Ukrainian tongue twisters cl | Symbolic Representations | Supervised | Text | PH |
| (Kovaliuk et al., 2024) | S & SA | Ukrainian tongue twisters cl | Symbolic Representations | Supervised | Speech | PH |
| (Bonafos et al., 2023) | S & MP | human vowel cl | Multi-Modal | Supervised | Speech | PH |
| (Bonafos et al., 2024) | S & MP | infant vocalization cl | Multi-Modal | Unsupervised | Speech | PH |
| (Gonzalez-Diaz et al., 2019; Paluzo-Hidalgo et al., 2022) | S & MP | emotion recognition | Multi-Modal | Supervised | Speech | PH |
| (Tlachac et al., 2020) | S & MP | depression detection | Multi-Modal | Supervised | Speech | PH |
| (Zhu et al., 2024) | S & MP | consonants recognition | Multi-Modal | Supervised | Speech | PH |
| (Sassone et al., 2022) | S & MP | music genre cl | Multi-Modal | Supervised | Speech | PH |
| (Vu et al., 2025) | S & MP | multi-modal adversarial robustness assessment | Multi-Modal | Supervised | Text | PH |
| (Bergomi, 2015) | S & MP | music cl | Multi-Modal | Supervised | Speech | PH |

Table 3: Non-theoretical applications of TDA in NLP. For the TDA techniques - PH: Persistent Homology and M: Mapper. Task categories - cl: classification, C & TM: clustering & topic modeling, S & SA: sentiment & semantic analysis, S & V: structure & visualization, H,S,& SA: health, social, & scholarly analysis, S & MP: speech & music processing, MI & A: model interpretation & analysis.

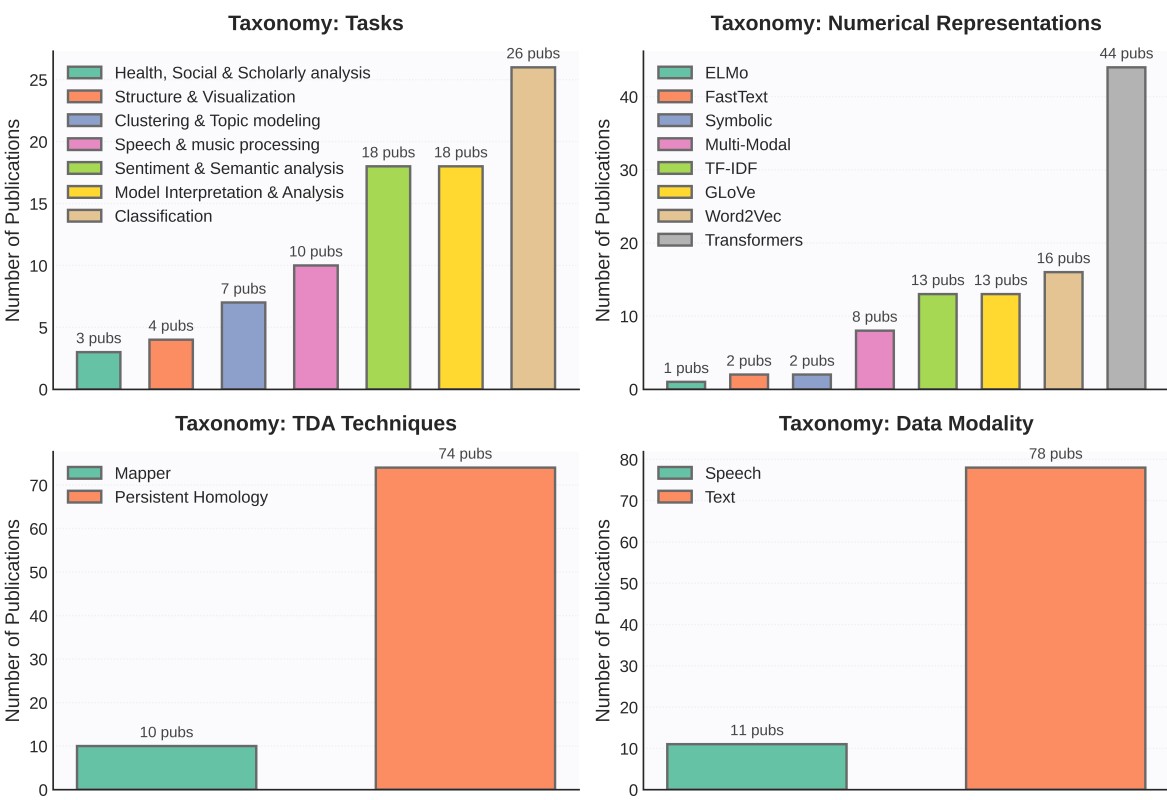

Figure 7: Number of publications in the sub-categories for the non-theoretical Taxonomies - Numerical Representations, Tasks, Modalities, and TDA techniques categories.

3. **Sentiment and Semantic Analysis**: The most popular applications are linguistic/grammatical acceptability (Cherniavskii et al., 2022; Jain et al., 2024), word sense induction and disambiguation (Rawson et al., 2022; Temčinas, 2018), and polysemy word classification (Jakubowski et al., 2020; Shehu, 2024).
4. **Structure and Visualization**: The most popular is using Mapper to visualize model hidden weights (Garcia, 2022; Rathore et al., 2023).
5. **Health, Social, and Scholarly Analysis**: Since this is not a popular application for TDA, the most interesting applications are - prediction of epidemics (Petri & Leitao, 2020) and categorization of lonely people (Effah, 2017).
6. **Speech and Music Processing**: The most popular applications are studying vocalizations (Bonafos et al., 2023; 2024) and music classification (Bergomi, 2015; Sassone et al., 2022).
7. **Model Interpretation and Analysis**: The most popular applications are model probing to reveal behavior in hidden weights (Kostenok et al., 2023; Gourgoulias et al., 2024).

> **Categories (Tasks).** *The seven categories of Tasks are selected based on surveying the types of problems researchers have attempted to solve by employing TDA techniques.*

## 5.2 Other Taxonomies: Learning Types, Modality, and TDA techniques

Finally, we can also categorize non-theoretical TDA applications in NLP by *learning types*, *Modality*, and *TDA techniques*. *Learning types* has supervised (Elyasi & Moghadam, 2019; Lavery et al., 2024), and unsupervised (Spannaus et al., 2024; Bonafos et al., 2024); *Modality* has text (Triki, 2021; Kostenok et al., 2023), and Speech (Sassone et al., 2022); and *TDA techniques*, has Persistent Homology (Torres-Tramón et al., 2015; Cherniavskii et al., 2022), and Mapper (Holmes, 2020; Elyasi & Moghadam, 2019).

Lastly, see Figure 7 for the number of publications for each sub-category of the four taxonomies - Numerical Representation, Task, Modality, and TDA techniques.

> **Categories (Learning Types, Modality, & TDA techniques).** *Due to the binary nature of these three categories, they are not very informative and thus unsuitable as the main taxonomy for the non-theoretical applications.*

## 5.3 Numerical Representation

### 5.3.1 TF-IDF

TF-IDF (Term Frequency - Inverse Document Frequency) is a well-known statistical formula that calculates the importance of words relative to a corpus. A few works investigated the extraction of topological features from TF-IDF representations as part of the pipeline illustrated in Fig. 6. For instance, SIFT, a persistent homology-based model with TF-IDF, is developed to differentiate between child and adolescent writings (Zhu, 2013). This model represented the TF-IDF features as a time series, and then extracted topological features to enhance text classification. Several other researchers applied this model to other *classification task*, such as deepfake text detection (Løvlie, 2023), presidential election speech attribution (Huang, 2022), distinguishing between languages by averaging the persistence landscapes Sovdat (2016), age group categorization of lonely people (Effah, 2017), and movie genre classification (Doshi & Zadrozny, 2018; Shin, 2019). Additionally, Elyasi & Moghadam (2019) compares the two popular TDA approaches - Persistent Homology and Mapper to classify Persian poems. Also, using Mapper for the *structure & visualization tasks*, Maadarani & Hajra (2020) explains linguistic phenomena in poetry writing styles, and van Veen (2020) interprets NLP model behavior. Lastly, we observe applications in the *clustering & topic modeling task* - keyphrase extraction (Guan et al., 2016), text summarization (Kumar & Sarkar, 2022), and twitter topic detection (Torres-Tramón et al., 2015); *sentiment & semantic analysis task* - legal entailment (Savle et al., 2019), and sentiment analysis of movie reviews (Michel et al., 2017).

> **Insight (TFIDF).** *Topological features extracted from TF-IDF have been applied to four out of the seven tasks - classification, clustering & topic modeling, sentiment & semantic analysis, and structure & visualization. This suggests that while there have been an exponential improvement in contextual and non-contextual word embeddings for numerically representing texts, TF-IDF can still extract decent features from text that can be further leveraged by TDA.*

### 5.3.2 Pre-trained Non-contextual Embeddings

**Word2Vec Embeddings.** Word2Vec embeddings are a type of word representation that allows words with similar meanings to have similar vector representations (Mikolov et al., 2013). Thus, we observe applications in the *structure & visualization task*, where Haghighatkhah et al. (2022) creates story trees to trace story lines. Next, we observe applications in *sentiment & semantic analysis task*, where TDA is applied to novel problems such as the creation of a topological search engine using Mapper (Cornell, 2020), measuring distance between the literary style of Spanish poets - Francisco de Quevedo, Luis de Góngora, and Lope de Vega (Paluzo Hidalgo et al., 2019), detecting narrative shifts in media discourse (Bailey & Heiligman, 2025), analysis of contradictions within texts (Wu et al., 2022), and word sense induction and disambiguation (Rawson et al., 2022; Temčinas, 2018). Thirdly, for the *classification* task, researchers detect fraudulent papers (Tymochko et al., 2021), and topological loops in logical statements (Tymochko et al., 2020).

Furthermore, we observe applications in the *health, social, and scholarly analysis* task - disease prediction from epidemic curves (Petri & Leitao, 2020), *topic modeling tasks* (Holmes, 2020; Wright & Zheng, 2020). Finally, for the *model interpretation and analysis* task, Feng et al. (2024) uses both topological and geometrical features to investigate the strength of LLM-enhanced data augmentation, Sun & Nelson (2023) derives the correlation between sentence vectors and their semantics, and Yessenbayev & Kozhirbayev (2022; 2024) compares the semantic alignment of text and speech embeddings.

> **Insight (Word2Vec).** *Topological features extracted from Word2vec are considered rich, such that researchers have attempted six out of seven tasks. The only task that have not been attempted are speech and music processing, since different embedding is needed for audio data. This suggests that word2vec embeddings capture enough linguistic features that when augmented with topological features can improve baseline performance. Finally, we observe that the sentiment and semantic analysis task is the most popular task for word2vec.*

**GloVe Embeddings.** GloVe or Global Vectors for Word Representation is another technique for numerically representing texts as embeddings. Topological features extracted from GloVe embeddings have been applied to the following tasks - *classification task*, which include author attribution of novel authors (Gholizadeh et al., 2018), fake news detection (Deng & Duzhin, 2022), and deepfake text detection (Løvlie, 2023); *sentiment & semantic analysis task*, which include document categorization (Gholizadeh et al., 2020), and capturing circles in circular arguments (Tymochko et al., 2020; Zadrozny, 2021b); *model interpretation and analysis*, where Haim Meirom & Bobrowski (2022); Michel et al. (2017) compare text representations & embeddings, Spannaus et al. (2024) explains model performance, and Zadrozny (2021a) tests the manifestation of intelligence and understanding in models; and *health, social, and scholarly analysis task* - social anxiety detection (Byers, 2021), and keywords extraction of scholarly documents (Novak, 2019).

> **Insight (GloVe).** *Topological features extracted from GloVe are considered rich, such that researchers have attempted all tasks, except (1) speech & music processing, (2) structure & visualization, and (3) clustering & topic modeling. The most popular application is in the model interpretation & analysis task, suggesting that glove embeddings can be sufficiently probed using TDA techniques to excavate interpretations of model performance.*

**FastText Embeddings.** FastText embeddings build on the Word2Vec approach by incorporating subword information, improving the representation of rare words, and allowing for embedding out-of-vocabulary words (Bojanowski et al., 2017). This type of embedding is not a popular feature extractor researchers employ to enhance topological features as only two tasks are attempted - *sentiment & semantic analysis task*, which include polysemy word classification (Jakubowski et al., 2020; Triki, 2021; Shehu, 2024), and word sense induction & disambiguation (Jakubowski et al., 2020); and *text classification*, where Tymochko et al. (2021) detects fraudulent papers.

> **Insight (FastText).** *FastText is not widely adopted for topological applications, with only a few researchers applying it to two tasks - text classification and sentiment & semantic analysis. Sentiment & Semantic analysis is the more popular application involving polysemy word classification and word sense induction & disambiguation.*

### 5.3.3 Pre-trained Contextual Embeddings

**Transformer Embeddings.** Researchers have evaluated the strength of the TDA features extracted from Transformer-based (Vaswani et al., 2017) embeddings. Using the idea of self-attention, the neural network can encode more semantic and syntactic features than previous embeddings which should allow for richer TDA features to be extracted. To incorporate TDA features for various tasks, several researchers have investigated the efficacy of using other outputs of encoder and decoder Transformer models - *CLS Embedding output*, *hidden weights*, and *attention weights* to extract high-quality additional features.

***CLS Embedding Output.*** Researchers have applied these features on *text classification task*, specifically for deepfake text detection (Tulchinskii et al., 2024; Kushnareva et al., 2024; Wei et al., 2025; Guilinger et al., 2025), fake news detection (Lavery et al., 2024), and TEDtalk public speaking ratings classification (Das et al., 2021). Additionally, we observe applications to the *topic modeling task* (Byrne et al., 2022), and the *model interpretation & analysis task* (Gourgoulias et al., 2024; Proskura & Zaytsev, 2024). Gourgoulias et al. (2024) probes LLMs to estimate class separability of text datasets, and Proskura & Zaytsev (2024) uses topological information from encoder models to select the best models to create an ensemble. Furthermore, Arun et al. (2025) employs TDA for the *structure &*

*semantic task*, detecting controversial vs. non-controversial political discourse by capturing the shifts in discourse for controversial data. Finally, Rathore et al. (2023) performs the *structure & visualization task* in combination with the *model interpretation & analysis task* by visualizing the training process of transformer-based models.

> **Insight (Transformer-CLS).** *CLS embedding is applied to five tasks - text classification (the most popular), topic modeling, model interpretation & analysis, sentiment & semantic analysis, and structure & visualization. The most interesting subtasks are probing LLMs to estimate class separability of text datasets and visualizing the training process of transformer-based models.*

**Hidden Weights.** *Classification task* include deepfake text detection (Uchendu et al., 2024), and LLM hallucination detection (Bazarova et al., 2025). Next, Garcia (2022) explore a combination of the *sentiment & semantic analysis* and *structure & visualization* tasks by using Mapper to visualize polysemous words in the hidden representations of the BERT transformer model. In addition, Bensalem et al. (2025), performs a *sentiment analysis* task by using the sentiment scores of original vs. translated texts extracted from a Transformer-based model, representing these scores in a time series form and using zigzag persistent homology to detect sentiment shift in translated texts. Similarly, Alexander & Wang (2023) combines the *health analysis* and *visualization* tasks to visualize GPT-3's embeddings of hate speech, misinformation, and psychiatric disorder texts with Mapper. Also, Ruppik et al. (2024) performs the *clustering & topic modeling task* through dialogue term extraction. Finally, we observe that the *model interpretation & analysis task* is the most popular task, where Gardinazzi et al. (2024) proposes a novel metric - *persistence similarity* to prune redundant layers in LLMs; Balderas et al. (2025) proposes Persistent BERT Compression and Explainability (PBCE) to compress BERT by pruning redundant layers; Sun & Nelson (2023) probes the correlation between sentence vectors and their semantics; Chauhan & Kaul (2022) proposes a novel scoring metric - *persistence scoring function* which captures the homology of the hidden representations of BERT; Fay et al. (2025) investigates the differences in the topological structure of the latent space of adversarial vs. non-adversarial texts in LLMs; García-Castellanos et al. (2024) performs zero-shot model stitching by employing *topological densification*; Fitz (2023) investigates the topological structure of the brain of ChatGPT concerning its notion of fairness; and Kudriashov et al. (2024) probes BERT's hidden weights on new grammatical features, known as *polypersonality*.

> **Insight (Transformer-Hidden).** *Researchers have applied TDA features extracted from Hidden weights in to all tasks, except speech & music processing tasks. The most popular task attempted is the model interpretation & analysis task, suggesting that TDA features extracted from the Hidden weights are able to make these black-box models more transparent.*

**Attention Weights.** Attention weights extracted from BERT and its variants (i.e., RoBERTa) have been transformed to both directed and undirected graphs, on top of which different TDA features are extracted for the *text classification task* such as deepfake text detection (Kushnareva et al., 2021), robustness evaluation of TDA features (Perez & Reinauer, 2022), authorship attribution of Japanese texts (Sakurai et al., 2025), out-of-distribution detection (OOD) (Pollano et al., 2024; Perez & Reinauer, 2022), and vulnerability detection in code (Snopov & Golubinskiy, 2024). Additionally, this framework is applied to the *sentiment & semantic analysis task*, specifically on human linguistic competence (i.e., grammatical acceptability judgment) (Cherniavskii et al., 2022; Proskurina et al., 2023; Perez & Reinauer, 2022), dialog term extraction (Vukovic et al., 2022), and document coherence (Jain et al., 2024). Similarly, with the same framework, we observe a *speech & music processing* application (Tulchinskii et al., 2023). Finally, researchers attempt the *model interpretation & analysis task*, where Kostenok et al. (2023) uses the topological features extracted from the attention weights to estimate uncertainty in the encoder models, and Proskura & Zaytsev (2024) performs dynamic weighting for building ensemble models.

> **Insight (Transformer-Attention).** *Given the wealth of information contained in the attention weights, researchers have applied topological features extracted from these weights to several tasks. The most interesting applications are observed on the sentiment and semantic tasks - grammatical acceptability judgment and document coherence analysis. Finally, we observe that for all the Transformer-based weights, the most explored subtask is deepfake text detection.*

**ELMo Embeddings.** ELMo embeddings are a type of word representation that captures both the meaning of words and their usage in context (Peters et al., 2018). Similar to other embeddings, ELMo has also been leveraged to extract topological features. Tymochko et al. (2021) performs *text classification* by applying this embedding for detecting fraudulent papers by examining their titles and abstracts. This is done in comparison of other embeddings (Word2Vec, GloVe, FastText, and Frequency Time Series) to ascertain the best embeddings to extract strong topological features.

> **Insight (ELMo).** *ELMo embeddings is applied to one task - text classification by one researcher in comparison to other embeddings to determine the best numerical representation for extracting rich TDA features.*

### 5.3.4 Symbolic Representations

Symbolic representations in the context of AI and cognitive science refer to the use of symbols such as letters, numbers, tokens, or abstract entities to represent concepts, objects, relationships, and rules within a system. These symbols can be manipulated according to predefined rules to perform reasoning, problem-solving, and decision-making. Symbolic representation contrasts with sub-symbolic representations, such as neural network-based embeddings, which do not explicitly use symbols or rules. This section then discusses the creation of symbolic representations by using *principles of letter coding (PLC)* and *principles of speech sound coding (PSSC)* which topological features are then extracted from.

PLC refers to rules and methods used to encode letters that fuel various communication systems, cryptography techniques, or linguistic analyses. Letter coding transforms letters or characters into different symbols, numbers, or other forms. PSSC is similar to PLC but for extracting topological features from speech sounds. One particular application of PLC and PSSC is the study of Ukrainian tongue twisters (Yurchuk & Gurnik, 2023; Kovaliuk et al., 2024). These applications attempt two tasks, *sentiment & semantic analysis* and *speech processing*, respectively.

Yurchuk & Gurnik (2023) uses the PLC to create word embeddings for Ukrainian tongue twisters and extract topological features from such embeddings with persistent homology. This is to distinguish tongue twister from a simple narrative sentence using SVM and decision tree classifiers. Similarly, Kovaliuk et al. (2024) use PSSC for classifying spoken Ukrainian tongue twisters.

> **Insight (Symbolic).** *Symbolic representations are a novel technique for numerically representing text. The only applications of this technique are on Uranian tongue twisters, both for the sentiment & semantic analysis and speech & music processing tasks. This suggests that such a technique could be applied to processing low-resource languages, where current popular numerical representation may be insufficient.*

### 5.3.5 Multi-Modal Representations

TDA features have also been extracted from other representations of NLP-related features, including multimedia data such as audio and video. In this section, the only task performed by researchers is speech & music processing, where applications include - studying human vowels and infant vocalizations (Bonafos et al., 2023; 2024), emotion recognition from audio speech (Gonzalez-Diaz et al., 2019) and audio in videos (Paluzo-Hidalgo et al., 2022), depression detection from audio clips (Tlachac et al., 2020), recognizing voiced and voiceless consonants in speech (Zhu et al., 2024), music classification (Bergomi, 2015; Sassone et al., 2022), and assessing the adversarial robustness of image-text multi-modal models by measuring topological consistency (Vu et al., 2025).

> **Insight (Multi-Modal).** *Multi-modal representations are another novel technique for numerically representing. The only task performed by researchers using this embedding is the speech & music processing task. This is because of data modality - audio data. All researchers show that topological features extracted from such embeddings experience high gains. Finally, the most interesting applications are the emotion recognition from audio speech and depression detection from audio clips.*

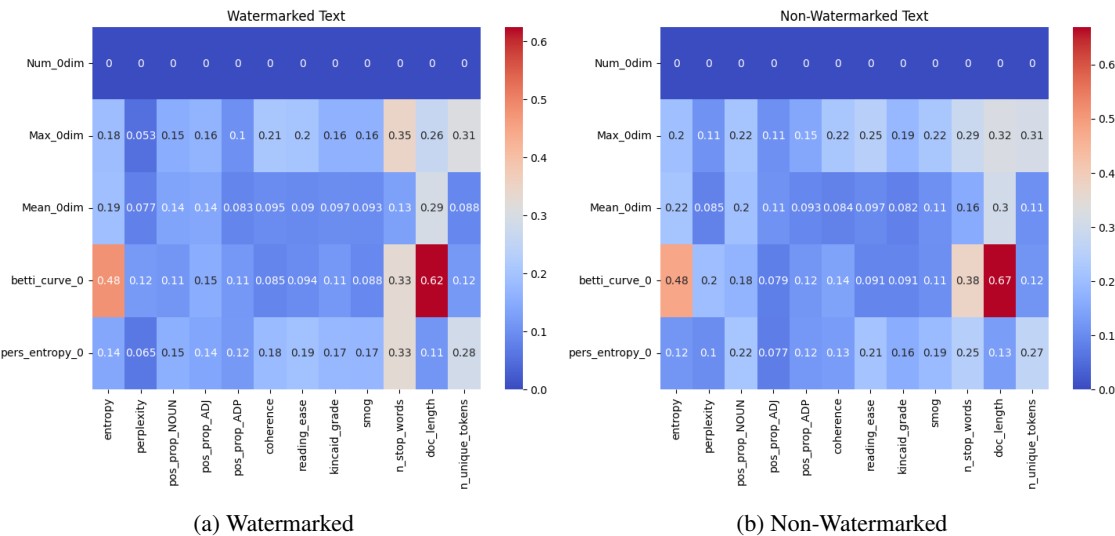

(a) Watermarked                    (b) Non-Watermarked

Figure 8: Using Watermarked vs. Non-Watermarked Texts, (a) & (b) are the distance correlation matrix of TDA features vs. Linguistic features heatmaps

# 6  Case Studies of TDA for NLP Tasks

This section showcases the utility of TDA in non-trivial NLP tasks by applying persistent homology and mapper using both *Theoretical* (Section 6.1) and *Non-theoretical* (Section 6.2) approaches.

## 6.1  Theoretical Approaches

**(Semantic) Topological Space.**   One of the many applications of TDA in NLP includes projecting the semantic knowledge from texts into a topological space, creating a *semantic topological space*. While, this framework has yielded interesting findings as seen in Section 4.1.1, we are still unable to intuitively explain how and which semantic features TDA captures. To mitigate this, we aim to answer the question - *What semantic features does TDA capture?* Additionally, by attempting to answer this question, we illustrate a real-world application of TDA in NLP. We perform this analysis on a relevant topic - distinguishing between watermarked and non-watermarked LLM-generated texts.

We extract topological and linguistic features for a correlation test. This means that if the features are correlated, it could suggest that these TDA features extracted from text capture semantic features. For the analysis, we use a subset of the C4 dataset that has watermarked and non-watermarked LLM-generated texts[4]. Next, we use the $ripser$ python package to extract the topological features with persistent homology and $textdescriptives$ Python package[5] to extract the linguistic features. The text is initially represented numerically with BERT's attention embeddings, then as an undirected graph, similar to Kushnareva et al. (2021) to extract topological features. Next, using the undirected graph as input, we calculate topological features in the 0-dimension, such as *the number of non-zero values, maximum value, mean of values, number of values*, and  *persistent entropy*. Also, for the linguistic features, we extract entropy, perplexity, position of noun & adjective, coherence, flesh reading ease score & grade, number of stopwords, length of text, and number of unique words. Finally, we use distance correlation to measure the statistical dependence of the two non-linear sets of features. See Figures 8a & 8b for the correlation matrix heatmap. Many of the TDA features correlate with the linguistic features, with the highest correlation being the length of text (doc_length) and number of topological values in $\beta_0$ (betti_curve_0). Lastly, these results suggest that topological features can capture semantic features, however, a more comprehensive study is needed to make this a strong finding. In addition, such a finding can inform the non-theoretical approaches we adopt for real-world NLP tasks.

---

[4]https://huggingface.co/datasets/acmc/watermarked_c4_dataset
[5]https://github.com/HLasse/TextDescriptives?tab=readme-ov-file

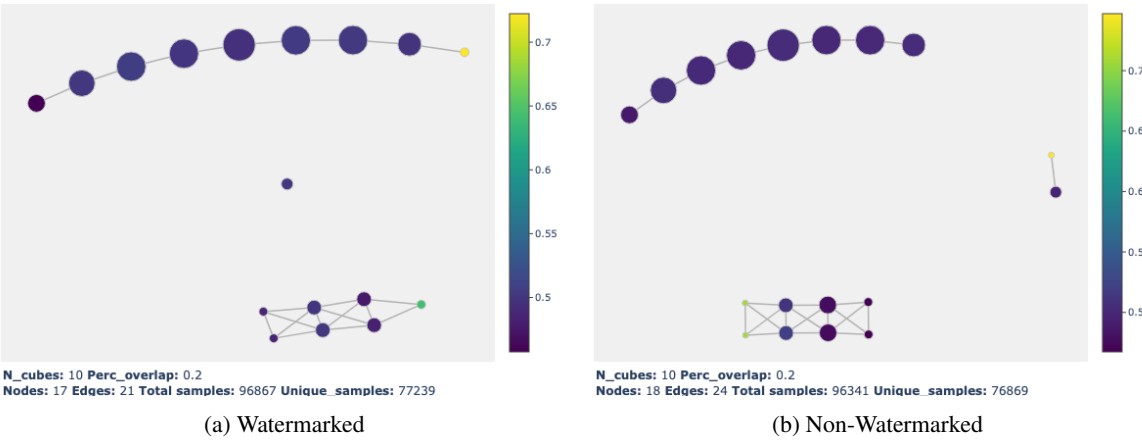

N_cubes: 10 **Perc_overlap:** 0.2
**Nodes:** 17 **Edges:** 21 **Total samples:** 96867 **Unique_samples:** 77239

N_cubes: 10 **Perc_overlap:** 0.2
**Nodes:** 18 **Edges:** 24 **Total samples:** 96341 **Unique_samples:** 76869

(a) Watermarked                    (b) Non-Watermarked

Figure 9: Using Watermarked vs. Non-Watermarked Texts, (a) & (b) are Mapper plots to visualize the shape of data.

**Topological "Shape" of Words**   Using the same method above to represent the Watermarked vs. Non-Watermarked texts, we then visualize the "shape" of their data using Mapper to observe the similarity or distinctness between the text types. Using the same C4 dataset, we create Mapper diagrams using the $Mapper$ Python package[6]. Figures 9a and 9b depict the Mapper plots of the Watermarked and Non-Watermarked texts, respectively. Represented as intuitive structured graphs, the figures show that texts across the two categories are topologically similar, although the watermarked text has slightly more nodes and edges than the non-watermarked texts, suggesting more connectivity. Thus, we can conclude that these texts are topologically similar, which could suggest that the watermarking technique does not significantly deviate from the style and quality of the non-watermarked version. In other words, such findings could indicate that the watermarking technique does not break the semantics, style, and syntax of the non-watermarked pre-trained model. However, to further explain these results, a more comprehensive study is required to interpret topological features extracted from text linguistically. This further stresses the use of the theoretical study of TDA as a powerful analytical tool as described in Section 6.1.

## 6.2   Non-Theoretical Approaches

**Model Interpretation and Analysis.**   To illustrate an interesting task for applying TDA to NLP, we aim to investigate whether the embeddings of English and non-English text pairs possess similar topological structures. This analysis can provide insights into the ability of text embeddings to capture similar structural information across different languages. The idea is to represent an English text with a monolingual English embedding model and a non-English text with the same meaning as a multilingual embedding model, and compare the similarities between these embeddings. This similarity is captured by calculating both the Euclidean and topolog-

| Distance | Value (Avg $\pm$ std) |
|---|---|
| Euclidean | $1.412 \pm 0.0249$ |
| Wasserstein | $0.054 \pm 0.0204$ |

Table 4: Average Euclidean and Wasserstein distances and standard deviation of the English-French text pairs

ical distances. For the topological distance, we use Wasserstein distance, a similarity metric between two probability distributions, typically used in computational topology (Panaretos & Zemel, 2019; Shin, 2019; Draganov & Skiena, 2024). Specifically, we evaluate the Euclidean and Wasserstein distances between the embeddings of English-French translation text pairs[7] (using the first 500 text pairs). For the Euclidean distance, we compare the embeddings from the English model and the multilingual model using the popular *Sentence Transformer* family encoders[8]. To calculate the Wasserstein distance, we first extract topological features from the embeddings, represent them as persistent diagrams, and then calculate the distance. Euclidean and Wasserstein distance range is $[0, \infty)$. We further normalize the embeddings such that both distance metrics are comparable.

---

[6]https://github.com/scikit-tda/kepler-mapper/tree/master
[7]https://huggingface.co/datasets/aircrypto/English-French-Translations
[8]https://huggingface.co/sentence-transformers/

| Pros | Cons |
|---|---|
| **Captures Global & Local Text Structure** – TDA models relationships between words, sentences, or documents in a topological space, uncovering hierarchical and contextual structures. | **High Computational Cost** – Applying TDA to large text corpora requires significant processing power, especially when constructing high-dimensional topological features. |
| **Robust to Noise & Variability** – Persistent homology can filter out minor textual variations while retaining essential linguistic structures. | **Interpretability Challenges** – Persistence diagrams and barcodes are not intuitive for NLP practitioners, requiring additional processing to extract meaningful linguistic insights. |
| **Effective for Low-Resource Settings** – Unlike deep learning, which requires large datasets, TDA can work with smaller corpora by leveraging topological structures rather than statistical frequency-based methods. | **Limited Software & NLP-Specific Tools** – Most TDA tools are designed for point clouds and image data, requiring adaptations for NLP applications. |
| **Detects Complex Relationships** – TDA can uncover semantic relationships and linguistic patterns that traditional word embeddings may miss. | **Not Yet Standard in NLP Pipelines** – TDA is still experimental in NLP, lacking standardized frameworks for integration with common NLP libraries like spaCy, Transformers, or NLTK. |
| **No Need for Explicit Feature Engineering** – Unlike many traditional methods, TDA analyzes raw data without requiring predefined features. | **Limited Adoption** – TDA is still an emerging field, meaning fewer case studies and industrial applications in NLP compared to traditional methods. |
| **Works Well with Word Embeddings & Transformers** – Can be integrated with word2vec, and Encoder & Decoder model embeddings to enhance understanding of text structures and improve classification tasks. | **Requires Specialized Expertise** – Implementing TDA in NLP requires both topology and NLP expertise, making adoption difficult for standard NLP practitioners. |
| **Provides Geometric Insights into Syntax & Semantics** – Helps visualize the shape of linguistic structures, which is valuable in understanding complex texts. | **Difficult to Benchmark** – Unlike traditional NLP metrics (e.g., BLEU, perplexity), there is no clear evaluation standard for TDA-based NLP models. |

Table 5: Pros and Cons of Topological Data Analysis (TDA) in NLP

Table 4 shows an average distance of 1.412 and 0.054 for the Euclidean and Wasserstein distances, respectively, both of which are relatively low values. These values suggest that these embeddings are able to capture the similarity between the text pairs, with Wasserstein being almost zero, demonstrating that the text pair embeddings are topologically similar, although one is multilingual and one is monolingual. This also suggests that TDA techniques can be used to evaluate the strength of multilingual models by observing their topological similarity to each individual monolingual embeddings.

# 7 When to use TDA in NLP

**Quick decision rules**

- Try TDA when multi-scale shape or connectivity of an embedding point-cloud (neighborhoods, loops, components) plausibly encodes a signal that the usual similarity pipelines miss (e.g., polysemy or author classification).
- Do not reach for TDA as a first pass for large end-to-end classification problems where massive supervised models already dominate unless you plan to use TDA as an orthogonal feature, regularizer, or diagnostic; gains are typically complementary, not a wholesale replacement.
- Use TDA analysis for explainability (what shape does an embedding produce?) or as a targeted augmentation (topological summaries + standard features), not as a black-box substitute for representation learning.

**Contexts where TDA is likely to help**

- *Word-sense, polysemy detection, or sense induction:* Local neighborhood topology around a word can reflect multiple senses (multiple connected components). Jakubowski et al. (2020) show topological measures of polysemy correlate with sense counts and produce competitive sense-induction results.
- *Authorship or stylistic signature (small-to-moderate corpora):* Authors may induce distinct global connectivity patterns across entity- or function-word clouds; persistent homology has been used as document-level topological signatures for novelist classification.
- *Storyline, discourse structure or topic evolution:* Mapping sentences or paragraphs into embeddings and tracking multi-scale connectivity can reveal hierarchical or story structure (e.g., "story trees").
- *Data augmentation or generative-data assessment:* TDA can quantify whether augmented/generated examples occupy the same topological region as original data (useful for judging augmentation quality). Recent work uses TDA to compare LLM-augmented vs. embedding-similarity augmentation.
- *Robustness, anomaly, or singularity detection in embeddings:* Topology-based measures (e.g., persistent H0, H1 summaries) can detect singularities or anomalous neighborhoods (useful for debugging or filtering embeddings).
- *Representation regularization or topology-preserving learning:* Adding topological losses (or using topological autoencoders) helps preserve connectivity and global shape in learned latent spaces — helpful when preserving manifold structure matters.

**Contexts where TDA is unlikely to help**

- Very large-scale classification where you can train huge supervised models and compute resources are limited
- Problems where pairwise similarity or local clustering fully captures the signal
- When you must extract features in real-time on resource-limited hardware

## 8  Discussion

### 8.1  Pros and Cons of TDA Applications in NLP

There are several benefits and challenges in adopting TDA techniques to solve non-trivial problems. We outline these pros and cons for various NLP problems in Table 5. The most glaring benefit and challenge are that TDA is effective in low-resource settings and the high computational cost associated with employing TDA, respectively. This benefit makes TDA particularly attractive for problems with noisy, insufficient, heterogeneous, and high-dimensional datasets. Beyond this, TDA can capture both global and local text structures, uncover complex relationships, and is robust to noise. However, issues such as interpretability, limited software support, and the need for specialized expertise still limit its widespread adoption in NLP. Although the high computational cost presents challenges for the adoption of TDA, recent advancement and improved accessibility of computational resources provide some relief. Section 9 will discuss how researchers can leverage these benefits to address the open problems and future directions.

### 8.2  Other non-TDA Topological Approaches Applied in NLP

There are other topological approaches for extracting structural information from data. Similar to TDA, these approaches also borrow from the mathematical field of algebraic topology, utilizing some of TDA's concepts including simplicial homology and Morse theory. However, they are not as rigorous as Persistent Homology.

Previous works have utilized such techniques in assessing document coherency by using the connected component dimension ($\beta_0$), representing the semantic topological space (Chiang, 2007). Similarly, Santacana (2025) uses a similar framework to build a framework for topological dialogue semantics in the Wolfram Language (i.e., a high-level, symbolic programming language). Next, Ionescu et al. (2025) proposes a framework known as *generative topolinguistics*, which projects texts to a topological semantic space to understand sociolinguistic phenomena in LLMs. Additionally, to extract the shape of text, Luong et al. (2007), considers text as a topological space by capturing the topology of linguistic features in texts. Which is used to identify the important stylistic features unique to specific authors or genres, specifically the frequent verbs that can be found in literature Luong et al. (2007).

From these applications, it is evident that all applications adopt the theoretical approaches used to reveal and confirm linguistic phenomena. This suggest the importance of topological methods in probing language to find confirmation of known historical linguistic knowledge and insights. Techniques such as this are useful on novel tasks such as

discovering the similarities between high-resource languages and low-resource languages, which could potentially help save some of these dying languages.

## 8.3 Insights into Interesting Applications of TDA in NLP

**Theoretical Approaches.** We observe that the most popular application of TDA in NLP has been confirmation of the topologically relationship between language families. For instance, Port et al. (2018; 2022) using the framework of a *syntactic topological space* and Draganov & Skiena (2024); Dong (2024), *topological "shape" of words*, they observe and confirm relationships between different language families. In the *syntactic topological space*, the aim is to capture the syntactic structure of language families from a topological perspective, such that languages in the same family have similar structures. For example, Port et al. (2022) finds that the syntactic topological structure of the Indo-European language family confirms the historical linguistic phenomena that the Hellenic branch played a role in its development. Next, for the *topological "shape" of words*, the aim is to capture the topological shape of these languages, such that languages of the same family have similar shapes. Moreover, Draganov & Skiena (2024)'s topological shapes of Indo-European languages confirm Port et al. (2018)'s findings.

Thus, by applying TDA to such tasks, we may find new linguistic phenomena which can inform how we numerically represent texts in the future, such that it captures all relevant linguistic principles.

**Non-Theoretical Approaches.** We suggest that the most interesting application of the non-theoretical approaches of TDA in NLP is in the *model interpretation & analysis* task. Researchers particularly utilize TDA features extracted from numerical representations of texts (i.e., mostly Transformer-based embeddings) to analyze LLM outputs in order to make them less opaque. See the subtasks explored under the *model interpretation and analysis* task:

1. Comparing text representations & embeddings (Haim Meirom & Bobrowski, 2022)
2. Explaining model performance (Spannaus et al., 2024)
3. Testing the manifestation of intelligence and understanding in models (Zadrozny, 2021a)
4. Dynamic weighting for building ensemble models (Proskura & Zaytsev, 2024)
5. Probing LLMs to estimate class separability of text datasets (Gourgoulias et al., 2024)
6. Visualizing the training process of transformer-based models (Rathore et al., 2023)
7. Model compression by pruning redundant layers (Gardinazzi et al., 2024; Balderas et al., 2025)
8. Probing the correlation between sentence vectors and their semantics (Sun & Nelson, 2023)
9. Capturing the homology of hidden representations of BERT (Chauhan & Kaul, 2022)
10. Zero-shot model stitching (i.e., Topological densification) (García-Castellanos et al., 2024)
11. Investigating the topological structure of how ChatGPT makes fair and unfair decisions (Fitz, 2023)
12. Probing BERT's hidden weights on new grammatical features (i.e., Polypersonality) (Kudriashov et al., 2024)
13. Uncertainty estimation (Kostenok et al., 2023)
14. Comparing the alignment of text and speech embeddings (Yessenbayev & Kozhirbayev, 2022; 2024)
15. Investigating the latent space representation of adversarial vs. non-adversarial texts in LLMs (Fay et al., 2025)
16. Investigating the geometry of textual data augmentation (Feng et al., 2024)

These subtasks illustrate novel ways in which researchers use TDA to probe the hidden weights of AI models in order to make them less opaque. This suggests that while TDA features are not intuitively explainable, it can be used to probe model weights such that model performance is interpretable.

## 8.4 Differences between standard NLP techniques and TDA

### 8.4.1 Qualitative differences

**Sensitivity to global structure:** Standard embeddings (i.e., Word2Vec, BERT) primarily encode local co-occurrence or contextual similarity. TDA (via persistent homology) captures global and multi-scale structure — e.g., loops, voids, or connected components in the embedding space. For instance, in semantic drift analysis, TDA can detect loops corresponding to cyclical usage of terms over time (i.e., "cloud" in meteorology vs. computing), which embeddings alone treat as proximity without structure.

**Topological invariants vs. vector similarity:** Embeddings compare objects with cosine similarity or Euclidean distance. TDA computes persistence diagrams/barcodes that remain invariant under continuous deformations (scaling, rotation). For instance, clusters of synonyms may look similar in embedding space, but TDA can reveal whether the cluster is contractible (one component) or contains holes reflecting polysemy.

### 8.4.2 Quantitative differences

**Dimensionality reduction robustness:** In experiments, persistent homology applied to embeddings yields stable Betti curves that remain informative even after strong dimensionality reduction — unlike raw embeddings, which often degrade in discriminative power.

**Additional discriminative signal:** Studies (Hofer et al., 2020) show that adding persistence summaries to standard embedding features improves NLP classification tasks (sentiment, authorship). This suggests that TDA captures information orthogonal to embeddings — otherwise, no performance gain would occur.

**Polysemy detection:** Embeddings often "average out" multiple senses of a word. Persistent homology can quantify the multiplicity of connected components or loops in local neighborhoods, providing measurable evidence of polysemy (Jakubowski et al., 2020).

## 9 Open Problems and Future Directions

We discuss the open problems and future directions for TDA applications in NLP, as well as ways in which researchers can leverage these benefits and mitigate these risks.

### 9.1 LLMs to Convert TDA Concepts to Codes

One of the major challenges in applying TDA to NLP tasks is the steep learning curve associated with its mathematical foundations, which are often accessible only to expert audiences. Moreover, theorists who develop and understand these advanced concepts do not always collaborate with computational scientists to translate them into executable code. To address this gap, Liu et al. (2024) proposes leveraging ChatGPT to generate Python code for TDA concepts by training it on these mathematical foundations. Their findings suggest that ChatGPT can alleviate this bottleneck, particularly for complex TDA concepts like hypergraphs, digraphs, and persistent harmonic space, which have not been as heavily explored as the Vietoris-Rips complex (Liu et al., 2024). Similarly, experts can develop specialized code generators, such as fine-tuning models like Code-Llama (Roziere et al., 2023) on TDA concepts. Creating a dedicated LLM for TDA code generation could significantly lower the entry barrier, encouraging the NLP community to explore TDA applications more innovatively.

### 9.2 Theoretical Approaches Connecting TDA Features with Linguistic Principles

There is a need for theoretical approaches that better tie TDA features to linguistic phenomena. For instance, Draganov & Skiena (2024) investigates the shape of words and their embeddings in Indo-European languages and finds similar conclusions as Port et al. (2018) and Port et al. (2022), who investigate the syntactic topological space of such languages. They find that TDA features represent historical facts, such that languages clustered closely together are similar or influenced by each other. Finally, we observe only 11 theoretical TDA works in NLP, compared to over 80 non-theoretical ones. Thus, we need more theoretical TDA approaches, as the depth and understanding of performance from a topological perspective, without further investigation, requires interpretation.

### 9.3 TDA for Interpretability

Interpreting TDA features in NLP problems, given its non-intuitive nature is very challenging. This is evident in the fact that most TDA for explainability applications is mostly in Computer Vision (Saul & Arendt, 2018), where the structure is distinct. Consequently, the interpretation of TDA features for text or speech data remains an open problem. There are currently two main tasks in this space - (1) explain model performance by interpreting TDA features; (2) explain model performance by using TDA to probe the prediction space or data. Either tasks require a

deeper understanding of TDA such that intuitive explanations can be used to tie topology to linguistic phenomena. Specifically, we need novel approaches that link TDA features to linguistic phenomena, for instance, disentangling $\beta_0$, and $\beta_1$'s representations to different properties of natural texts such as coherency, and writing style. This can be done either through visualizing the prediction space of a model (Rathore et al., 2023) or probing the prediction space of models with TDA (Gardinazzi et al., 2024; Xenopoulos et al., 2022; Solunke et al., 2024).

### 9.4 Novel Applications of TDA

When we have more theoretical approaches of TDA and issues barring the application of TDA on interpretable NLP tasks are mitigated, we can hope that TDA can be applied to even more novel, diverse and important tasks. From Section 5.1, we can see that TDA has been applied to 7 categories on non-theoretical NLP tasks. While many of the tasks are interesting, especially the speech & music processing and health applications, there are still nuanced niche fields that could benefit from TDA. One glaring application is on multi-lingual tasks, just as Haim Meirom & Bobrowski (2022) and García-Castellanos et al. (2024). Due to the benefits of TDA which include performing robustly on heterogeneous, imbalanced, and noisy data, its application on multi-lingual tasks is necessary. Other applications include: Topology-aware neural networks, Topological interpretability, semantic and syntactic structure, forensic authorship, embedding space, and multi-modal (e.g., language-vision model) analysis.

### 9.5 Improvement in TDA Feature Extraction

Unlike some other data modalities that have a distinct shape, texts take the shape of their numerical representation. However, different numerical representations capture different linguistic features, making it challenging to intuitively interpret their topological differences. Therefore, we must find novel ways of representing texts numerically, for example, the use of *symbolic representations* or better ways to use the numerical representations that exist in such a way that is advantageous for extracting the best TDA features.

### 9.6 Adversarial Robustness of TDA Features

Robustness to noise, particularly adversarial perturbations, has been an important research topic in NLP. While such robustness of TDA features is promising, there have been only a few works in this direction (Perez & Reinauer, 2022; Chauhan & Kaul, 2022). For instance, Perez & Reinauer (2022) show that their topologically-augmented BERT model is much more robust than the base BERT model when tested against perturbations generated by TextAttack (Morris et al., 2020). Chauhan & Kaul (2022) also shows that there are some weak correlations between persistent homology features of a trained BERT model and its adversarial robustness against several state-of-the-art attackers. However, all existing works only evaluate on BERT model with simple attack mechanism, missing other security scenarios such as poisoned or backdoor attacks. Therefore, we call for a more comprehensive robustness evaluation of TDA on NLP models, especially from the NLP, ML, and security communities.

### 9.7 Topological Deep Learning for NLP

Due to the benefits of TDA and deep learning, a new niche field is born - Topological Deep Learning (TDL) a "the collection of ideas and methods related to the use of topological concepts in deep learning" (Papamarkou et al., 2024). Initially, TDL is described as an ensemble of topological features extracted by TDA techniques such as persistent homology and deep learning features. In this setting, TDL is a traditional deep learning model with extra features (i.e., TDA-extracted features). However, as the field has advanced, a new definition for TDL has emerged - "the collection of ideas and methods related to the use of topological concepts in deep learning" (Papamarkou et al., 2024). TDL allows a deep learning model to be integrated more deeply with concepts of algebraic topology, such as the introduction of simplicial neural networks (NNs), which are NNs with layers made up of simplicial complexes. This deeper integration of TDA into NNs makes TDL particularly useful for the explosion of high-dimensional data. These high-dimensional data need better tools for processing as the current tools shrink the dimension, resulting in information loss. In NLP, one particular approach to integrate TDA with high-dimensional NLP embeddings has been the utilization of text in graphical forms, which have been shown to yield better results than directly using texts as a sequence of tokens (Zhong et al., 2020; Liu et al., 2023; Phan et al., 2023). Nevertheless, more research is still needed to validate such an approach.

## 10 Conclusion

Our world is currently experiencing an explosion of data and an explosion of computational techniques to process such data. Machine Learning (ML) is the most popular of these computational methods; however, while its benefits are numerous, it has a few limitations. The biggest of the limitations of ML is its inability to sufficiently process data that is high-dimensional, imbalanced, noisy, and scarce. Therefore, a small community of NLP researchers emerged to tackle this limitation by proposing using TDA to tackle difficult NLP tasks. These researchers employ two TDA techniques - Persistent Homology and Mapper to solve NLP tasks using theoretical and non-theoretical approaches. This yielded 100 papers, which we comprehensively surveyed in this paper. Finally, we conclude that while the applications of TDA in NLP have improved greatly since 2013, there is still room for improvement, specifically in reducing the barrier to entry for non-TDA experts to apply it to their NLP tasks.

## 11 Ethical Statement

This survey highlights emerging applications of Topological Data Analysis (TDA) in Natural Language Processing. While our primary goal is to synthesize existing work, we recognize that several use cases carry important ethical considerations and dual-use risks. Topological methods can inadvertently expose latent sensitive attributes (e.g., dialect, health cues, authorship), enabling re-identification or profiling even when data is anonymized. Techniques discussed, such as watermarking, could also be repurposed to circumvent provenance systems. Applications in speech, emotion, and health-related domains further raise fairness, consent, and equity concerns, particularly for minority groups and low-resource languages. Therefore, we emphasize the need for bias and robustness audits, careful data governance and licensing, and privacy-preserving mechanisms when sharing derived features. Responsible release practices—such as restricting code that enables circumvention, conducting red-team evaluations, and requiring IRB or ethics review for clinical or surveillance-adjacent uses—are essential. Finally, given the computational demands of TDA pipelines, their environmental impact should be considered as well.

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
