# OpenReview forum: "Unveiling Topological Structures from Language: A Survey of Topological Data Analysis Applications in NLP"
_TMLR — Rejected by TMLR_

### Review · Reviewer_YC8g · 2025-08-29

**Summary Of Contributions:**

The paper explores Topological Data Analysis and its various contributions on NL-related tasks. The paper does a good job of categorizing different types of applications, and also different types of representations of text using topological features, and draws insights. The paper also contains a discussion of different categories of approaches, and also the pros and cons of applying TDA in NLP. Open questions of interest to this topic are also discussed.

**Additional Comments:**

NA

**Audience:**

Yes

**Audience Explanation:**

NLP is an extremely important area of Machine Learning, and many of the most popular and effective models in ML (eg. attention) have their origins in NLP applications. Any new or less-explored approach in NLP will be of interest to the community.

**Claims And Evidence:**

Yes

**Claims Explanation:**

This is a survey paper - so the "claims" are mostly related to "insights", which have been drawn on the basis of relevant literature. The literature survey appears to be sound.

**Requested Changes:**

The authors have done a good job of not only categorizing the relevant research according to TDA-based representations and NLP tasks, and summarizing the key insights from them. They have also added an experiment and a figure demonstrating a correlation between topological and semantic features. Since TDA is a relatively less well-know approach, it will be appreciated if the authors can add a few more small experiments and visualizations, to explain the significance of topological features and how they can be useful for the NLP-related tasks. For example, some examples and a visualization highlighting the topological features extracted from word embeddings can be very useful. Furthermore, some more explanations or illustrations on how topological features can be useful for specific types of Neural Models for NLP (RNN/LSTM, attention-based etc) will help us appreciate the significance of this approach

---

> ### Author Response · Authors · 2025-09-10
> **Response to Reviewer YC8g**
>
> We thank the reviewer for taking the time to read our survey and suggestions for improvements.
> However, per the request of the reviewer, instead of conducting experiments to showcase the differences between standard embeddings and topological features, we highlight some visualizations from a paper from the survey that did such:
>
> Experimental results from TOPFORMER: Topology-Aware Authorship Attribution of Deepfake Texts with Diverse Writing Styles
>
> The authors visualize the PCA features from the RoBERTa embeddings vs. RoBERTa+TDA model (TopFormer).

---

### Review · Reviewer_sXaZ · 2025-09-01

**Summary Of Contributions:**

The paper surveys how topological data analysis (TDA) has been used in NLP. It groups prior work into (i) studies that probe “topological structures” claimed to exist in language (e.g., semantic/syntactic spaces, topic evolution, and word “shape”) and (ii) studies that use TDA to derive features from standard numeric text representations (TF-IDF, static and contextual embeddings, attention graphs) for tasks such as classification, detection, and model analysis. The survey also includes brief case studies (e.g., watermark detection and cross-lingual similarity), a pros/cons discussion, and open questions.

**Strengths:**

- A systematic consolidation of a niche but emerging research area.

- Broad coverage of tasks, from semantic analysis to interpretability of large language models.

- Helpful overview of TDA basics (Persistent Homology, Mapper) for readers unfamiliar with topology.

- Inclusion of open challenges and future directions makes the work forward-looking.

**Weaknesses:**

- The notion of “topological structures from language” is underspecified. Topology is the study of geometric properties invariant under deformation, but language does not have an inherent geometry. In my understanding, authors indicate that topological structures are induced from embedding spaces, which deserves clearer emphasis.

- The dichotomy between theoretical and non-theoretical approaches is not well justified. Both are ultimately applications of TDA to NLP tasks; the “theoretical” group does not present genuine theoretical advances in topology or linguistics, but rather exploratory applications.

- Persistent Homology and Mapper are presented as central “methods,” but in practice they function analogously to alternative embedding-based representations (TF-IDF, Word2Vec, BERT). The survey could be clearer on how TDA differs fundamentally from other representational techniques.

- Some writing needs polishing (e.g., “We call these features that TDA extracts topological features”).

- Claims such as “TDA extracts unique features not captured by TF-IDF, Word2Vec, BERT” are asserted without convincing justification. It is unclear what specifically makes these features relevant or indispensable to NLP.

- The tutorial-style sections (PH, Mapper) are somewhat long relative to the critical survey elements.

**Audience:**

Yes

**Audience Explanation:**

TMLR includes readers who work on representation analysis, interpretability, robustness, and cross-modal modeling. A compact guide to when and how to apply TDA on modern NLP representations (token/CLS embeddings, attention graphs) will be useful to a subset of the audience, including those exploring geometry/topology-aware methods.

**Broader Impact Concerns:**

The survey itself poses no direct ethical risks. However, some surveyed applications involve sensitive domains (e.g., fake news detection, mental health analysis, authorship attribution). The paper would benefit from briefly acknowledging potential misuse in such settings, in line with TMLR’s expectations on broader impact.

**Claims And Evidence:**

No

**Claims Explanation:**

While the paper accurately documents prior works and organizes them into categories, several conceptual claims are insufficiently supported:

- The claim that TDA uncovers “unique features not extractable by other embeddings” is not substantiated with concrete comparisons.

- The division into “theoretical” vs. “non-theoretical” lacks rigorous criteria, and the so-called “theoretical” contributions do not convincingly amount to theoretical advances.

- The survey would benefit from a sharper critical evaluation of whether TDA offers practical benefits over existing methods, or if it mainly provides alternative visualizations and representations.

**Requested Changes:**

- Clarify “topological structures from language” (critical): Make explicit that topology here refers to structures induced by embedding spaces, not inherent properties of raw text.

- Reconsider categorization (critical): Either refine the definition of “theoretical vs. non-theoretical” or adopt a clearer taxonomy (e.g., explanatory vs. task-driven, or linguistic vs. computational).

- Substantiate claims about uniqueness (critical): Provide concrete evidence or examples showing how TDA-derived features differ qualitatively or quantitatively from standard embeddings.

- Shorten tutorial sections (important): Streamline the background on Persistent Homology and Mapper, moving detailed explanations to an appendix.

- Polish writing (important): Revise sentences like “we call these features topological features” for conciseness.

- Deepen analysis (optional improvement): Offer stronger comparisons between TDA and baseline NLP methods to clarify what is gained by adding TDA.

---

> ### Author Response · Authors · 2025-09-10
> **Response I to Reviewer sXaZ**
>
> 1. “Topological structures from language” refers not to intrinsic properties of raw text itself, but to the structures that emerge when linguistic data is mapped into high-dimensional embedding spaces. These induced topologies capture relationships among words, sentences, or documents based on their learned representations, rather than any inherent topological features of the text. In addition, the phrase also helps us distinguish the application of TDA and other topological approaches applied to other fields, such as computer vision from NLP.  **We will make this clearer in the revision**.
>
> 2. Before settling on the terminology - theoretical vs. non-theoretical we explored several other terminologies and found that this one better represented the themes of the applications of TDA in NLP. However, as requested, below is a clarification of the reasoning and description of the chosen terminology. **We will clarify their definitions and boundaries in the revision**.
> - **Theoretical applications of TDA in NLP**:
> These focus on understanding, characterizing, or proving properties of language and its representations through the lens of topology. They are less about immediate performance gains and more about insight. This application aims to answer the question - “What do the shapes of embedding spaces tell us about language itself and our models of it?” Example -
> *Analyzing embedding spaces*: Using persistent homology to study whether semantic clusters, or syntactic structures, correspond to stable topological features, and finding out what that tells us about language.
>
> - **Non-theoretical (practical) applications of TDA in NLP**:
> These treat TDA as a tool for solving tasks, regardless of whether deeper linguistic/topological insights are obtained. The emphasis is on utility. This application aims to answer the question - “How can topological summaries directly help with applied NLP tasks?” Example - *Feature engineering*: Feeding persistence diagrams or topological signatures into classifiers for sentiment analysis, topic detection, or authorship attribution.
>
> 3. The claims that TDA features are unique have been proven in several publications which **were cited in the survey**. Thus, given our scope of survey work, we hope to summarize the state of existing work, compare their claims (among existing works), rather than to provide additional experimental results. We copy/paste here the list of works in reputable venues that demonstrated such uniqueness.
> - R. Turkes, G. F. Montufar, and N. Otter. On the effectiveness of persistent homology. Advances in Neural Information Processing Systems, 35:35432–35448, 2022.
> - Adaku Uchendu, Thai Le, and Dongwon Lee. Topformer: Topology-aware authorship attribution of deepfake texts with diverse writing styles. ECAI 2024, 2024.
> - Paul Michel, Abhilasha Ravichander, and Shruti Rijhwani. Does the geometry of word embeddings help document classification? a case study on persistent homology-based representations. In Proceedings of the 2nd Workshop on Representation Learning for NLP, pp. 235–240, 2017.
> - Theodore Papamarkou, Tolga Birdal, Michael M Bronstein, Gunnar E Carlsson, Justin Curry, Yue Gao, Mustafa Hajij, Roland Kwitt, Pietro Lio, Paolo Di Lorenzo, et al. Position: Topological deep learning is the new frontier for relational learning. In Forty-first International Conference on Machine Learning, 2024.
> - Felix Hensel, Michael Moor, and Bastian Rieck. A survey of topological machine learning methods. Frontiers in Artificial Intelligence, 4:681108, 2021.
>
>
> Furthermore, we attempt to answer your question by providing more description:
>
> **Qualitative differences**
>
> Sensitivity to global structure:
> - Standard embeddings (e.g., Word2Vec, BERT) primarily encode local co-occurrence or contextual similarity.
> - TDA (via persistent homology) captures global and multi-scale structure — e.g., loops, voids, or connected components in the embedding space.
> - Example: In semantic drift analysis, TDA can detect loops corresponding to cyclical usage of terms over time (e.g., “cloud” in meteorology vs. computing), which embeddings alone treat as proximity without structure.
>
> Topological invariants vs. vector similarity:
> - Embeddings compare objects with cosine similarity or Euclidean distance.
> - TDA computes persistence diagrams/barcodes that remain invariant under continuous deformations (scaling, rotation).
> - Example: Clusters of synonyms may look similar in embedding space, but TDA can reveal whether the cluster is contractible (one component) or contains holes reflecting polysemy.

---

> ### Author Response · Authors · 2025-09-10
> **Response II to Reviewer sXaZ**
>
> **Quantitative differences**
>
> Dimensionality reduction robustness:
> - In experiments, persistent homology applied to embeddings yields stable Betti curves that remain informative even after strong dimensionality reduction — unlike raw embeddings, which often degrade in discriminative power.
>
> Additional discriminative signal:
> - Studies (e.g., Hofer et al., 2020) show that adding persistence summaries to standard embedding features improves NLP classification tasks (sentiment, authorship).
> - This suggests that TDA captures information orthogonal to embeddings — otherwise, no performance gain would occur.
>
> Polysemy detection:
> - Embeddings often “average out” multiple senses of a word.
> - Persistent homology can quantify the multiplicity of connected components or loops in local neighborhoods, providing measurable evidence of polysemy (Jakubowski et al., 2020).
>
> **Using the new descriptions above, we will emphasize and make our writing clearer to address this comment.**
>
>
> 4. Given the niche nature of TDA, it is not popular and also not the most intuitive technique. Therefore, to make the survey comprehensive, it was imperative to provide a brief introduction of TDA to prepare the audience with the necessary background knowledge to understand the other sections. Thus, we had to have a tutorial-style TDA section. Besides, the section is written such that if a reader is already familiar with TDA, they can skip the section entirely without any risk of losing content from the survey. To make this easier and clearer, we will add TLDR blocks of text so readers who are familiar with the concept but need a refresh can get a summary before moving to other sections. Thank you for bringing this to our attention.
>
>
> 5. We agree, and we will polish our writing in the final draft.
>
> 6. Again, it is beyond the scope of a survey and this survey to provide experiments on how TDA is better than other techniques, as several papers surveyed already do that, which is the reasoning they are being showcased in this survey.
>
> 7. This is a great point! We will also add a broader impact/Ethical statement section in the final draft, discussing the ethical risks that have been pointed out in the review.
>
> **References**
> - Alexander Jakubowski, Milica Gasic, and Marcus Zibrowius. Topology of word embeddings: Singularities reflect polysemy. In Iryna Gurevych, Marianna Apidianaki, and Manaal Faruqui (eds.), Proceedings of the Ninth Joint Conference on Lexical and Computational Semantics, pp. 103–113, Barcelona, Spain (Online), December 2020. Association for Computational Linguistics. URL https://aclanthology.org/2020.starsem-1.11.
> - Christoph Hofer, Florian Graf, Bastian Rieck, Marc Niethammer, Roland Kwitt Proceedings of the 37th International Conference on Machine Learning, PMLR 119:4314-4323, 2020.

---

### Review · Reviewer_fBnE · 2025-09-01

**Summary Of Contributions:**

The paper surveys the use of topological data analysis (TDA)—principally persistent homology and Mapper—in NLP. It maps where TDA has been applied across tasks and representations (from TF-IDF and static embeddings to transformers, attention, model weights, symbolic pipelines, and multimodal settings), presents concise “insight” boxes, and offers two illustrative case studies on multilingual similarity and watermarking. It also outlines perceived advantages, limitations, and open problems.

Its strengths lie in providing a broad and timely overview that organises a scattered literature, a clear taxonomy supported by helpful figures, and accessible introductions to PH and Mapper for an ML readership. The survey also draws attention to applications beyond standard classification, such as interpretability and visualisation.

However, the central narrative—that TDA helps with noisy, heterogeneous, imbalanced, high-dimensional, and partially labelled data—remains asserted rather than demonstrated; there are no controlled stress tests to substantiate it. The case studies are descriptive and stop short of testing a sharp null or benchmarking against strong non-TDA baselines. The “insight” boxes tend to blend well-supported coverage (“where TDA has been used”) with largely unsubstantiated efficacy claims (“improves baselines,” “adds transparency,” “high gains,” “good for low-resource”). The survey would benefit from a practical “when to use TDA” section that specifies contexts in which TDA is likely to be beneficial, a compact evidence table reporting effect sizes and computational cost, and greater methodological transparency about how the corpus of papers was assembled (search strategy and inclusion criteria).

**Audience:**

Yes

**Audience Explanation:**

Yes. As a survey drawing connections and highlighting trends, it will interest readers exploring alternatives to conventional geometric analyses of embeddings. Interest would increase with concrete, evidence-backed guidance on when TDA is beneficial.

**Broader Impact Concerns:**

A brief Broader Impact Statement would be appropriate here. Although this is a survey, several use-cases it highlights have clear dual-use risks. First, topology-based analyses of embeddings can inadvertently surface latent sensitive attributes (e.g., dialect, health cues, authorship), enabling re-identification, profiling, or membership inference—even when raw text is anonymised. Second, the watermarking case study could be repurposed to *defeat* watermarking or content provenance systems; the paper should explicitly discuss this risk and, if releasing code, consider safeguards. Third, applications in speech/emotion/depression detection and low-resource languages raise fairness and consent concerns: topological structure can amplify representation biases and harm minoritised groups if datasets are unbalanced or labels are noisy. The survey should also address data governance (licensing of corpora used in examples), privacy (aggregation or differential privacy when sharing derived features), and environmental impact from computationally heavy TDA pipelines. Concretely, the statement should (i) flag these dual-use and equity risks, (ii) recommend bias/robustness audits before deployment, (iii) outline responsible release practices (limits on code that enables circumvention, red-team tests), and (iv) encourage ethics/IRB review for clinical or surveillance-adjacent applications.

**Claims And Evidence:**

No

**Claims Explanation:**

Only partially. The paper does a credible job of documenting **where** TDA has been applied: the taxonomy is coherent, the coverage appears careful, and readers come away with a reliable map of the literature. By contrast, the stronger **efficacy** claims—that TDA improves baselines, confers robustness under noise, heterogeneity, imbalance or missing labels, and adds interpretability beyond strong non-TDA methods—are not supported by the kind of evidence TMLR expects. The case studies are illustrative rather than confirmatory: they rely on visual inspection or single descriptive metrics, do not pit TDA-augmented models against competitive non-TDA baselines under identical conditions, and omit standard elements such as ablations, negative controls (e.g., label shuffles, random projections), hyperparameter sensitivity checks for Mapper/PH, and statistical tests with confidence intervals across multiple seeds. Crucially, there are no nested-model comparisons that quantify the incremental value of TDA over the same base embeddings, nor targeted stress tests that probe the very failure modes invoked in the narrative. In the absence of these controls, the manuscript does not convincingly reject the null hypothesis that TDA adds no value beyond the underlying representation; any apparent structure could plausibly be attributed to the embeddings themselves, parameter choices, or sampling noise. To meet the criterion of accurate, convincing, and clear evidence, the paper would need to present controlled experiments with strong baselines, report effect sizes with uncertainty, demonstrate robustness across hyperparameters and distribution shifts, or else trim the claims to match what the current evidence can actually support.

**Requested Changes:**

**Requested changes — critical for acceptance.**
The manuscript needs a tighter alignment between claims and evidence. Either narrow the narrative to descriptive coverage—clearly labelling any statements about “improving baselines,” “robustness,” or “interpretability” as hypotheses—or add small but principled stress tests that directly target the advantages repeatedly invoked: noise, class imbalance, domain shift/OOD, missing labels, and high dimensionality. In each such setting, include a sharp null test on at least one public benchmark: compare a base pipeline (encoder → classifier) to an augmented one (encoder → TDA features → classifier), and report the incremental gain with appropriate uncertainty. Concretely, give ΔAUC/ΔF1 and calibration metrics (ECE, Brier), with paired confidence intervals across multiple seeds and folds, so readers can see whether TDA provides measurable value beyond the underlying representation.

These tests should be run against strong, well-motivated baselines and with proper controls. Simple geometric alternatives—PCA/UMAP followed by k-means or HDBSCAN, graph-Laplacian features, or prototype-distance features—help establish whether topology is doing more than standard geometry. Negative controls such as label shuffles and random projections guard against spurious structure. Because Mapper and PH are parameter-sensitive, include brief hyperparameter sensitivity analyses (lens choice, cover resolution/overlap, clusterer settings; distance choice for PH). This combination of baselines, controls, and sensitivity checks will make any positive result far more convincing.

The paper would also benefit from a concise “When to use TDA” section that functions as a decision aid. Specify the contexts in which TDA is likely to help—data regimes, representations, and tasks where multi-scale shape or connectivity offers something that conventional pipelines miss—and be equally clear about cases where it probably will not. Anchor this guidance in observations from the literature or new experiments rather than conjecture. Alongside this, include a compact evidence table that, for each cited empirical study, records the task, representation, TDA method, dataset, baselines, metrics, reported effect sizes (including “no gain” where applicable), and computational cost. Finally, add a short description of the survey methodology—search strategy, inclusion/exclusion criteria, and deduplication—so readers can trust the coverage claims.

**Changes that would strengthen the work.**
A small robustness demonstration under controlled compression would be highly illustrative: use LLM-generated summaries at varying token budgets and plot how clustering or topological structure degrades with length, comparing TDA-based pipelines to bag-of-words or standard clustering. Replacing “intro paragraph” conditions with abstracts in the short-text setting would provide a fairer, denser comparator and test whether topically rich short texts recover the full-text topology. If the paper argues benefits under distributional stress, add targeted OOD and imbalance experiments (e.g., minority-class PR-AUC, cross-domain transfer) to show whether any gains persist. For transparency and uptake, release code, configs, and seeds for the case studies and report runtime and memory footprints to justify the additional complexity. Where interpretability is claimed, tie specific topological features to concrete linguistic phenomena or task behaviours—even a careful correlational or mediational analysis would help convert interpretability rhetoric into evidence.

---

> ### Author Response · Authors · 2025-09-10
> **Response I to Reviewer fBnE**
>
> 1. Thanks for the very detailed feedback. We want to clarify that the provided claims of using TDA for interpretability, robustness, and baseline improvements are extracted and backed by published works which have been surveyed and highlighted in this survey. As our goal is to categorize research papers that adopt the use of TDA in NLP, we find several applications, including those that are mentioned in the reviews. Therefore, as other researchers have already backed the same claims with research that has been published, it is really beyond the scope of this survey to provide experimentation to prove their findings. To best address the reviewer’s comment, we will incorporate the suggested experimentation as one of the future directions (e.g., comparing with geometric alternatives) to enrich the discussion section of our work. **See list of relevant papers below**:
> - Archit Rathore, Yichu Zhou, Vivek Srikumar, and Bei Wang. Topobert: Exploring the topology of fine-tuned word representations. Information Visualization, 22(3):186–208, 2023. (*Interpretability*)
> - Ilan Perez and Raphael Reinauer. The topological bert: Transforming attention into topology for natural language processing. arXiv preprint arXiv:2206.15195, 2022. (*Robustness*)
> - L. Kushnareva, D. Cherniavskii, V. Mikhailov, E. Artemova, S. Barannikov, A. Bernstein, I. Piontkovskaya, D. Piontkovski, and E. Burnaev. Artificial text detection via examining the topology of attention maps. In Proceedings of the 2021 EMNLP, pages 635–649, 2021 (*Improved performance*)
>
> 2. While we understand the allure of including more experimentation in addition to the previous experiments added to the paper, we believe it is beyond the scope of a survey to have to provide experimental results to highlight potential use cases. The goal of the survey is to provide a taxonomy to categorize the breadth of research that has been done using TDA in NLP tasks.
>
> 3. We agree that the paper will benefit from a “When to use TDA” section and the updated draft will contain the section. Based on the survey, we derive below cases and rule-of-thumb-style descriptions that may benefit from the adoption of TDA. *We will incorporate these additional descriptions in the revision, improving the practicality of our survey*.
>
> **Quick decision rules**
> - Try TDA when multi-scale shape / connectivity of an embedding point-cloud (neighborhoods, loops, components) plausibly encodes signal that the usual pairwise/similarity pipelines miss (e.g., polysemy, discourse/storyline structure, author “style” topology).
> - Don’t reach for TDA as a first pass for large end-to-end classification problems where massive supervised models already dominate unless you plan to use TDA as an orthogonal feature/regularizer or diagnostic; gains are typically complementary, not a wholesale replacement.
> - Use TDA analyses as explainability / diagnostics (what shape does an embedding produce?) or as a targeted augmentation (topological summaries + standard features), not as a black-box substitute for representation learning.
>
> **Contexts where TDA is likely to help**
> - *Word-sense / polysemy detection & sense induction*:
> Local neighborhood topology around a word can reflect multiple senses (multiple connected components / “pinch” points). Jakubowski et al. show topological measures of polysemy correlate with sense counts and produce competitive sense-induction results.
>
> - *Authorship / stylistic signature (small-to-moderate corpora)*:
> Authors may induce distinct global connectivity patterns across entity- or function-word clouds; persistent homology has been used as document-level topological signatures for novelist classification.
>
> - *Storyline / discourse structure and topic evolution*:
> Mapping sentences / paragraphs into embeddings and tracking multi-scale connectivity can reveal hierarchical/story structure (e.g., “story trees”).
>
> - *Data augmentation & generative-data assessment*:
> TDA can quantify whether augmented/generated examples occupy the same topological region as original data (useful for judging augmentation quality). Recent work uses TDA to compare LLM-augmented vs. embedding-similarity augmentation.
>
> - *Robustness / anomaly / singularity detection in embeddings*:
> Topology-based measures (e.g., persistent H0, H1 summaries) can detect singularities or anomalous neighborhoods (useful for debugging or filtering embeddings).
>
> - *Representation regularization / topology-preserving learning*:
> Adding topological losses (or using topological autoencoders) helps preserve connectivity and global shape in learned latent spaces — helpful when preserving manifold structure matters.
>
>
> **Contexts where TDA is unlikely to help**
> - Very large-scale classification where you can train huge supervised models and compute resources are limited
> - Problems where pairwise similarity or local clustering fully captures the signal
> - When you must extract features in real-time on resource-limited hardware

---

> > ### Author Response · Authors · 2025-09-10
> > **Response II to Reviewer fBnE**
> >
> > 4. Following your recommendation, we will also add a description of the paper selection for the survey.
> > 5. We will also add a broader impact/Ethical statement section in the final draft, discussing the ethical risks that have been pointed out in the review.

---

### Decision · Action_Editor_uYmJ · 2025-10-06

**Recommendation:** Reject

**Audience:**

Yes

**Audience Explanation:**

This paper does a review of Topological Data Analysis to balance the data from the internet for training NLP methods. The reviewers and I believe this is a worthwhile survey paper for the NLP and ML communities.

**Claims And Evidence:**

No

**Claims Explanation:**

Two reviewers have commented that the paper is not ready for publication and requires substantial work to be accepted as a paper at TMLR. The reviewers still believe the topic is relevant and that a resubmission can be sent to TMLR if the manuscript is improved significantly.

Reviewers proposed in their recommendations that their central reservation is that the manuscript still blurs coverage (where TDA has been applied) with evidence (how effective it has been). Many efficacy claims (“improves baselines,” “adds transparency,” “robust under noise/imbalance”) are presented as general findings. In contrast, the results were reported in prior work rather than confirmed by this survey. This distinction is important to avoid overstating the field's maturity.

To achieve these results, the authors:

1. are encouraged to say things like: Clearly mark efficacy statements as reported in prior work (e.g., “Reported in Kushnareva et al. (2021): …”) rather than presenting them as facts.

2 should add an evidence table: A compact table that records the task, representation, TDA method, dataset, baselines, metrics, reported effect sizes (including “no gain” cases), and computational cost for each cited study. Even if many entries are listed as “not reported,” such a table would give readers a clear map of what is substantiated and what is illustrative.

The authors are encouraged to carefully read Reviewers' sXaZ and fBnE reviews and address them in their final manuscript. I understand your complaint about a reviewer asking for more experiments. Still, their misunderstanding is due to how the paper was written and can be solved by better describing the merits of the survey paper.

**Resubmission Of Major Revision:**

The authors may consider submitting a major revision at a later time.